# Tgfβ signaling is critical for maintenance of the tendon cell fate

Guak-Kim Tan[1], Brian A Pryce[1], Anna Stabio[1], John V Brigande[2], ChaoJie Wang[3], Zheng Xia[3], Sara F Tufa[1], Douglas R Keene[1], Ronen Schweitzer[1,4]*

[1]Research Division, Shriners Hospital for Children, Portland, United States; [2]Oregon Hearing Research Center, Oregon Health & Science University, Portland, United States; [3]Computational Biology Program, Oregon Health & Science University, Portland, United States; [4]Department of Orthopedics, Oregon Health & Science University, Portland, United States

**Abstract** Studies of cell fate focus on specification, but little is known about maintenance of the differentiated state. In this study, we find that the mouse tendon cell fate requires continuous maintenance in vivo and identify an essential role for TGFβ signaling in maintenance of the tendon cell fate. To examine the role of TGFβ signaling in tenocyte function the TGFβ type II receptor (*Tgfbr2*) was targeted in the Scleraxis-expressing cell lineage using the *ScxCre* deletor. Tendon development was not disrupted in mutant embryos, but shortly after birth tenocytes lost differentiation markers and reverted to a more stem/progenitor state. Viral reintroduction of *Tgfbr2* to mutants prevented and even rescued tenocyte dedifferentiation suggesting a continuous and cell autonomous role for TGFβ signaling in cell fate maintenance. These results uncover the critical importance of molecular pathways that maintain the differentiated cell fate and a key role for TGFβ signaling in these processes.

*For correspondence:
schweitz@ohsu.edu

**Competing interests:** The authors declare that no competing interests exist.

## Introduction

Studies of cell fate determination are in most cases focused on the signaling pathways and transcription factors that direct naive cells to assume a specific cell fate (*Li et al., 2012*; *James, 2013*; *Huang et al., 2015*). It is commonly accepted that once fully differentiated the cells enter a stable cellular phenotype, but relatively little is known about the molecular mechanisms that reinforce and maintain this differentiated state. Maintenance of the differentiated state is, however, essential for tissue function and identifying the molecular pathways involved in these processes may be of great importance for understanding tissue homeostasis and pathology.

Tendons are connective tissues that transmit forces from muscle to bone to generate movement (*Kannus, 2000*). Despite their importance to overall musculoskeletal function and their slow and limited healing capabilities, relatively little is known about tendon development, the tendon cell fate, maturation and pathology. Elucidating the key molecular regulators of these processes is thus essential for improvements in the management of tendon healing, the treatment of tendinopathy and for bioengineering efforts for this tissue.

A limited number of transcription factors were so far identified as key regulators of the tendon cell fate including most notably, Scleraxis (*Scx*), a bHLH transcription factor expressed in tendon cells from progenitor stages and through development (*Schweitzer et al., 2001*) and Mohawk (*Mkx*), an atypical homeobox protein with essential roles in the development of the collagen matrix in tendons (*Ito et al., 2010*). Prototypic markers for the tendon cell fate also include the transmembrane protein tenomodulin (Tnmd) and collagen type I (*Kannus, 2000*; *Huang et al., 2015*), the major building blocks of the tendon fibrillar extracellular matrix that mediates the transmission of force by tendons.

Previous studies have also established a central role for the transforming growth factor-β (TGFβ) signaling pathway in early events of tendon development (*Pryce et al., 2009*; *Havis et al., 2016*). Notably, TGFβ is a potent inducer of *Scx* both in vivo and in cultured cells and disruption of TGFβ signaling in mouse limb bud mesenchyme resulted in complete failure of tendon formation (*Pryce et al., 2009*). This phenotype manifested at the onset of embryonic tendon development but robust expression of TGFβ ligands and associated molecules in later stages of tendon development suggested possible additional roles for TGFβ signaling in tendon development (*Kuo et al., 2008*; *Pryce et al., 2009*). Moreover, subcutaneous application of growth and differentiation factors (GDFs), members of the TGFβ superfamily, can induce ectopic neo-tendon formation in rats (*Wolfman et al., 1997*). The goal of this study was therefore to ask if TGFβ signaling plays essential roles at later stages of tendon development.

The TGFβ superfamily comprises secreted polypeptides that regulate diverse developmental processes ranging from cellular growth, differentiation and migration to tissue patterning and morphogenesis (*Santibañez et al., 2011*; *Sakaki-Yumoto et al., 2013*). These ligands act by binding to transmembrane type II receptors, which in turn recruit and activate a type I receptor. The activated receptor complex subsequently phosphorylates and activates receptor-regulated transcription factors called Smads (Smad2/3 for TGFβ signaling) that then complex with the common-mediator Smad4 and translocate into the nucleus where they promote or repress responsive target genes (*Vander Ark et al., 2018*). The TGFβ proper ligands (TGFβ1–3) all bind to a single type II receptor. Consequently, disrupting this one receptor is sufficient to abrogate all TGFβ signaling. To test for additional roles of TGFβ signaling in tendon development and biology, we wanted to bypass the early essential function in tendon formation, and decided to target TGFβ type II receptor (*Tgfbr2*) directly in tendon cells. We therefore targeted the receptor using *ScxCre* (*Blitz et al., 2013*), a tendon-specific Cre driver, so that TGFβ signaling will be disrupted specifically in tendon cells and only after the initial events of tendon formation.

We find that tendon differentiation function and growth during embryonic development was not disrupted following targeted deletion of TGFβ signaling in tenocytes, but shortly after birth the cells lost tendon cell differentiation markers and reverted to a more progenitor-like state. Moreover, viral reintroduction of *Tgfbr2* to mutant cells was sufficient to prevent dedifferentiation and even to rescue the tendon cell fate in a cell autonomous manner, highlighting a continuous and essential role of TGFβ signaling in maintenance of the tendon cell fate.

## Results

### Targeting TGFβ type II receptor in Scx-expressing cells resulted in tendon disruption and limb abduction

Our previous studies showed that disruption of TGFβ signaling in mouse limb mesenchyme resulted in the complete failure of tendon formation (*Pryce et al., 2009*). To examine later roles of TGFβ signaling in mouse tendon development, the floxed *Tgfbr2* gene was targeted conditionally with *ScxCre* (*Tgfbr2^{f/-};ScxCre*; called hereafter *Tgfbr2;ScxCre mutant*) to bypass the early role of TGFβ signaling in tendon development. *ScxCre* activity in tenocytes is not uniform during embryogenesis (*Figure 1—figure supplement 1A*) and complete targeting of tenocytes is achieved only in early postnatal stages. Indeed, immunostaining for TGFβ type II receptor revealed that by P0 mutant tendons displayed a nearly complete loss of receptor expression (*Figure 1—figure supplement 1C*). Consequently, *Tgfbr2;ScxCre* mutant embryos developed a complete network of tendons by E14.5, indicating they have bypassed the early requirement for TGFβ signaling in tendon development (*Figure 1A*).

Mutant tendon development was not perturbed through embryogenesis and mutant pups appeared normal at birth (*Figure 1C*). However, by day 3 after birth (P3), mutant pups showed physical abnormalities that manifested in abducted paws, splayed limbs (*Figure 1C*, black arrowhead) and severe movement limitations. Examination of forelimb tendons of P7 mutant pups using the tendon reporter *ScxGFP* revealed severe tendon disruptions. A few lateral limb tendons, for example the extensor carpi radialis longus tendon underwent fragmentation and disintegrated (*Figure 1B*, yellow arrowhead and *Figure 1—figure supplement 3*), whereas the majority of other tendons, notably the extensor digitorium communis tendons, retained structural integrity with a substantial

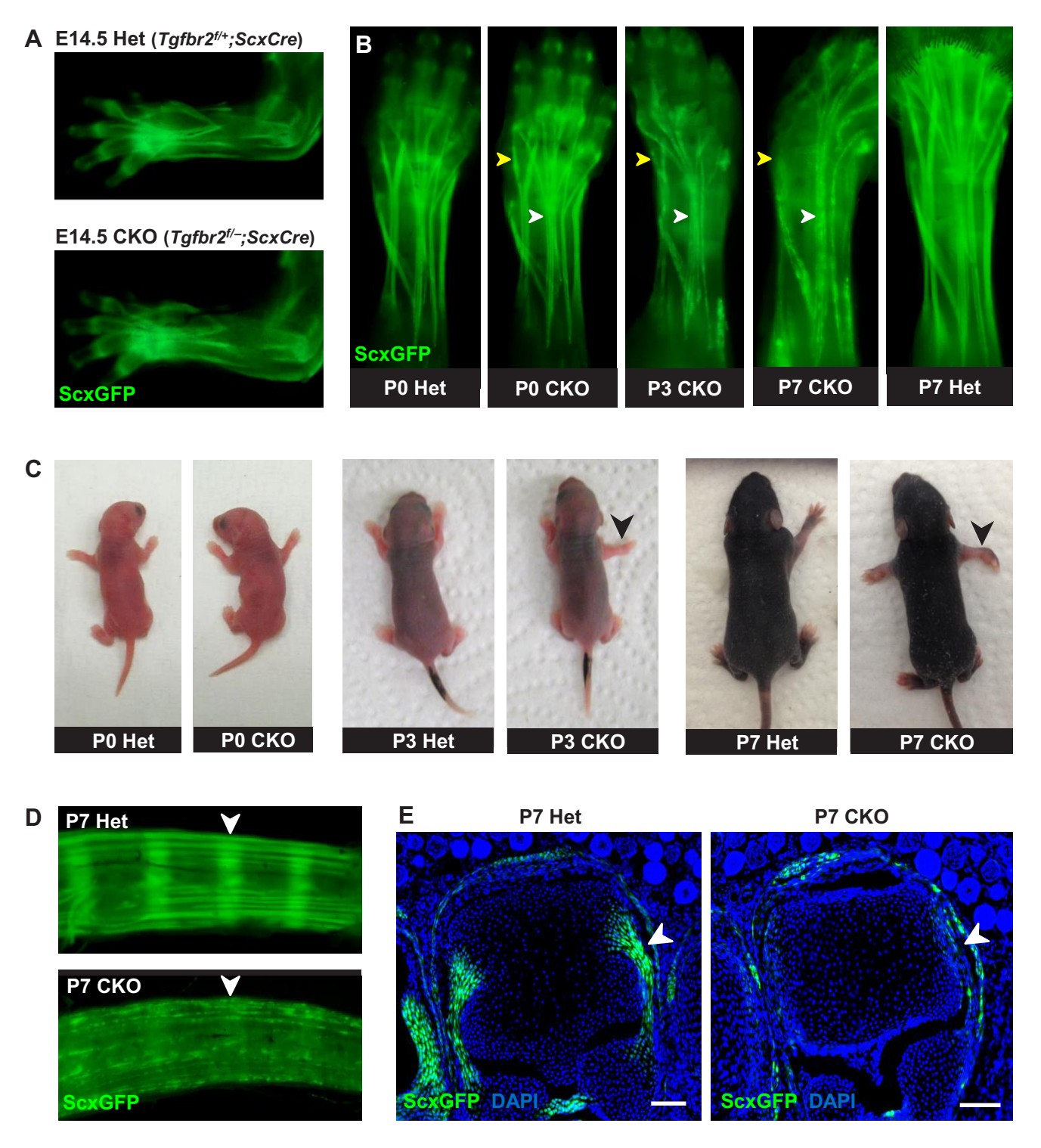

**Figure 1.** Tendon phenotypes manifested in *Tgfbr2;ScxCre* mutants. (A–D) Whole-mount imaging in fluorescent *ScxGFP* signal or brightfield. (A) Dorsally viewed embryo forelimb shows the formation of a complete network of tendons in both mutant and heterozygous control by E14.5. (B) Tendons of mutant pups appeared intact at birth, but by P3 lateral tendons disintegrated and were eventually eliminated (yellow arrowheads), whereas the majority of other tendons persisted with a substantial loss of the *ScxGFP* signal (white arrowheads). (C) Mutant pups appeared normal at birth but showed physical abnormalities including abducted paw and splayed limb (black arrowheads) by P3. (D–E) Substantial loss of *ScxGFP* signal was also detected in all tendons and related tissues. (D) Tail tendons and annulus fibrosus of the intervertebral disc (white arrowheads) in P7 pups. (E) Collateral

*Figure 1 continued on next page*

*Figure 1 continued*

ligaments of the metacarpophalangeal joint imaged in transverse section through the joints of heterozygous control and mutant pups at P7 (white arrowhead). Scale bar, 100 µm. Mutant: CKO, Heterozygous: Het.

The online version of this article includes the following figure supplement(s) for figure 1:

**Figure supplement 1.** Verification of the *Tgfbr2* knockdown efficiency in mutant cells.
**Figure supplement 2.** Gradual loss of tendon marker *ScxGFP* in *Tgfbr2;ScxCre* mutants at post-natal stages.
**Figure supplement 3.** Fragmentation and elimination of lateral tendons in *Tgfbr2;ScxCre* mutant neonates.
**Figure supplement 4.** Disruption of the flexor carpi radialis tendon in *Tgfbr2;ScxCre* mutant embryos.

loss of *ScxGFP* signal (*Figure 1B*, white arrowhead). Substantial loss of *ScxGFP* was also detected in all tendons and related tissues, including hindlimb and tail tendons, ligaments and the annulus fibrosus of the intervertebral disc (*Figure 1D,E*). Loss of *ScxGFP* signal was gradual starting around P2-P3, that is before the manifestation of physical abnormalities (*Figure 1B* and *Figure 1—figure supplement 2*). All mutant tendon cells lost *ScxGFP* at P7. We therefore performed most analyses of the mutant phenotype in this fully-phenotypic stage. The progressive nature of the phenotype also manifested in exacerbated movement limitations as mutant pups became older. This phenotypic progression was observed in most mutant pups but intriguingly, in rare cases (~2%) the mutant pups showed physical abnormalities and severe tendon phenotypes already at birth. Regardless, all mutants died at or before P14 likely due to *ScxCre* activity in developing cardiac valves (*Levay et al., 2008*), leading to enlarged heart as evidenced by gross examination and histological analysis (data not shown).

A closer examination of the mutant embryos identified the first indication of a tendon phenotype already at E16.5. The flexor carpi radialis tendons of mutant embryos were consistently torn by E16.5 (*Figure 1—figure supplement 4*). Interestingly, this phenotype was highly reproducible while the patterning and development of other tendons in mutant embryos was not perturbed through embryogenesis. Moreover, expression of the prototypic tenocyte markers *Scx*, tenomodulin and collagen I (*Figure 2A–D*) and the development of the collagen matrix were not disrupted in any tendon of mutant embryos (*Figure 2E,F*), including the flexor carpi radialis tendon before it snapped. A direct cause for the specific tear of the flexor carpi radialis tendon in mutant embryos was not identified to date.

Tendons are rich in collagen fibers that provide structural integrity to the tendons and transmit the forces generated by muscle contraction (*Kannus, 2000*). Since young mutant pups exhibited movement difficulties, we first examined possible structural effects in the collagen matrix. The ultrastructure of mutant tendons that remained intact was therefore analyzed by transmission electron microscopy (TEM). Surprisingly, despite the functional defects and loss of *ScxGFP* signal starting around P3, collagen fibers in mutant tendons appeared organized and indistinguishable from those of wild-type (WT) littermates at this stage (*Figure 3A,B*). Apparent collagen degradation was observed only in older mutant pups (≥P7) (*Figure 3C–G*), suggesting the disruption to the matrix of these tendons may be a secondary consequence of the cellular changes in these mutants and/or of their movement difficulties. Furthermore, epitenon, a monolayer of cells that engulf and define the boundary of the tendon (*Kannus, 2000*) (*Figure 3F*, black arrowhead), was gradually disrupted and in some cases was almost undetectable in older mutant pups (*Figure 3G*, white arrowhead), suggesting that loss of the tendon boundary is an additional feature of the phenotype in these mutants.

## Loss of the tendon cell fate in mutant tenocytes

As mentioned earlier, the *ScxGFP* signal in mutant tendons appeared patchy contrary to the smooth appearance of WT tendons (*Figure 1B*), suggesting a disruption at the cellular level. To examine this phenotype at the cellular level, we analyzed cross-sections through the extensor communis tendons of P7 WT and mutant pups. In P7 WT pups, all tendon cells were positive for *ScxGFP*, *Tnmd* and *Col1a1* (*Figure 4A,C*). Conversely, most cells in mutant tendons lost the *ScxGFP* signal and tendon marker gene expression (*Figure 4B*, white arrowhead and *Figure 4C*). Interestingly, some cells in mutant tendons retained *ScxGFP* signal and appeared rounded and enlarged from P3 onwards (*Figure 4B*, yellow arrowhead). Some of these cells exhibited weak or no expression of the *Ai14 Rosa26-tdTomato* (*RosaT*) Cre reporter (*Madisen et al., 2010*), suggesting a recent activation of the

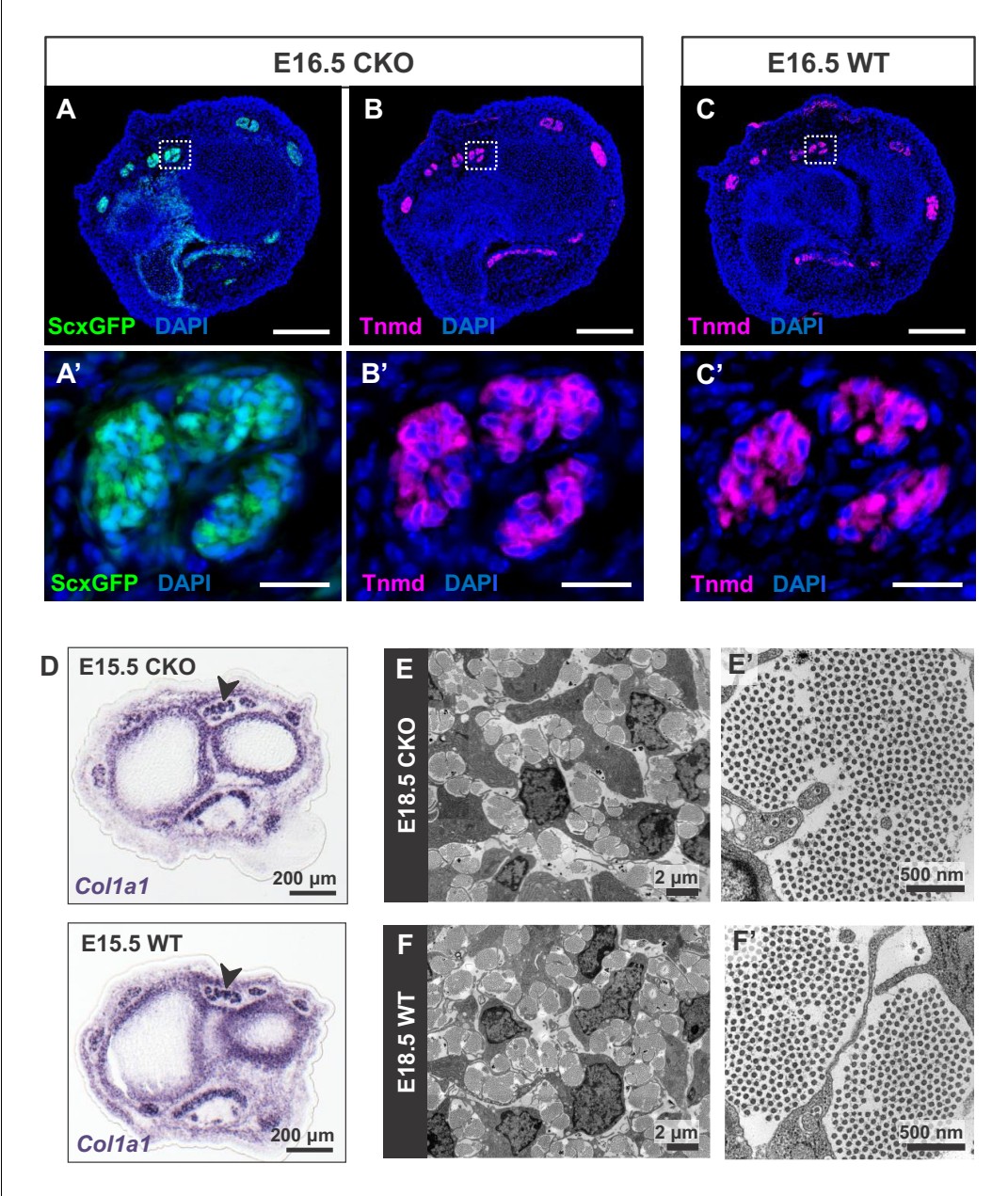

**Figure 2.** Tendon development in *Tgfbr2;ScxCre* mutant embryos was not perturbed through embryogenesis. (A) *ScxGFP* signal and (B) tenomodulin (Tnmd) immunofluorescence on transverse sections at wrist level of E16.5 mutant embryos demonstrate that the pattern and expression of prototypic tenocyte markers was not disrupted in mutant tendons. (C) Tnmd immunofluorescence in E16.5 wild-type tenocytes. (A'), (B') and (C') are higher magnifications of extensor digitorium communis tendons as boxed in (A), (B) and (C). (D) In situ hybridization for *Col1a1* on transverse sections of the forelimb from E15.5 mutant and wild-type littermates reveals that expression of the major matrix genes was not altered in mutant embryos (black arrowhead). (E,F) TEM images of tendons from forelimbs of E18.5 mutant and wild-type embryos reveals that organization and accumulation of the tendon extracellular matrix was not disrupted in the mutant. (E',F') Higher magnification views of (E) and (F) for direct visualization of the collagen fibers. Scale bars, 200 µm (A–C) and 20 µm (A'–C'). Mutant: CKO, Wild-type: WT.

The online version of this article includes the following figure supplement(s) for figure 2:

**Figure supplement 1.** Evaluating cell death, proliferation and transdifferentiation in *Tgfbr2;ScxCre* mutant tendons.

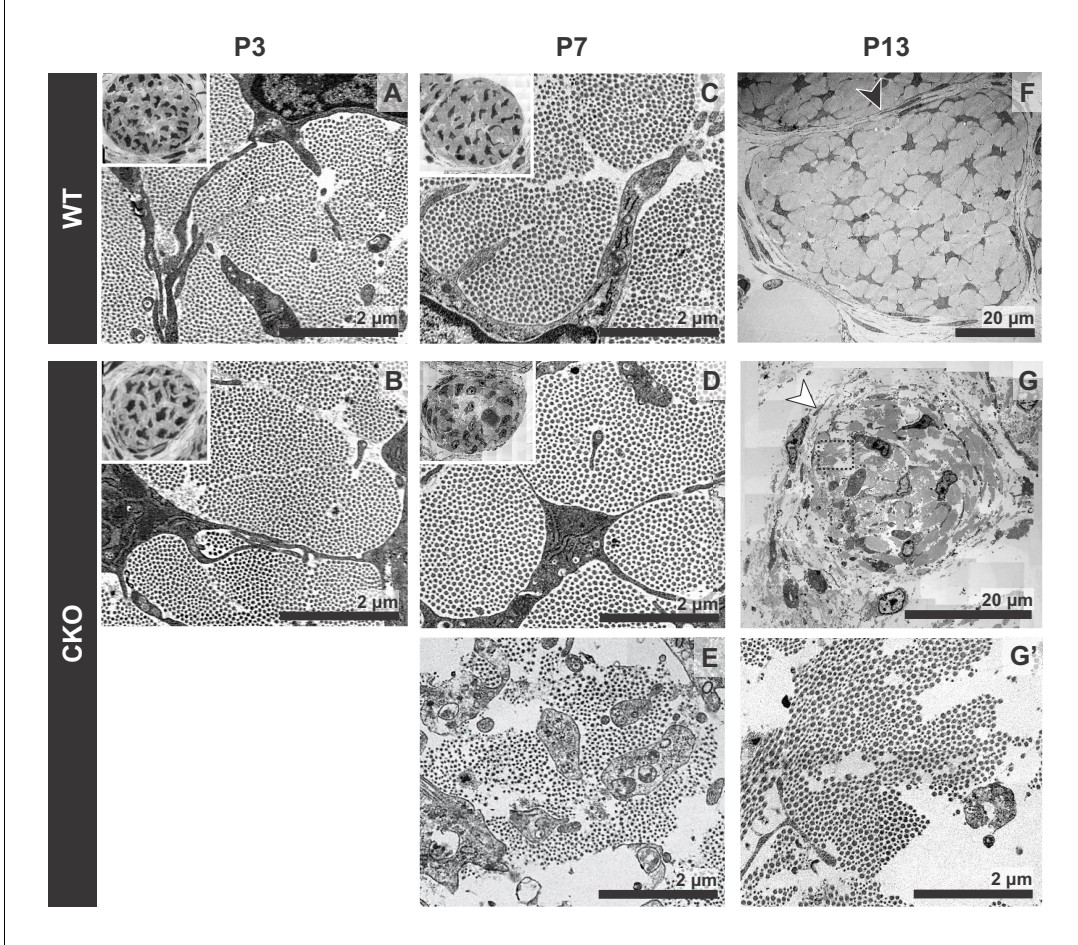

**Figure 3.** Tendon degeneration observed in *Tgfbr2;ScxCre* mutants only at later postnatal stages. TEM images of tendons from forelimbs of mutant and wild-type littermates at P3, P7 and P13. (**A,B**) Despite detectable functional defects starting around P3 in mutant pups, collagen matrix organization in mutant neonates was indistinguishable from that of their wild-type littermates. (**C–E**) By P7, the mutant tendon began to show signs of matrix degradation compared to the wild-type tendon. Collagen fibrils remained intact in some areas (**D**) and showed signs of deterioration in other areas (**E**). (**F,G**) Apparent collagen degradation and disrupted epitenon structures (white arrowhead) could be detected in tendons of P13 mutant pups. Black arrowhead indicates epitenon in wild-type pups. Boxed region in (**G**) is shown enlarged in (**G'**). Insets show transverse section TEM images of entire tendons at low-magnification (not to scale). Mutant: CKO, Wild-type: WT.

*Scx* enhancer in these cells and therefore that they are newly recruited tendon cells. Analysis of this aspect of the mutant phenotype will be published in a separate manuscript (Tan et al. in preparation).

The fact that most cells in the mutant tendons do not express tendon markers is surprising, since the cells in these tendons were functional tenocytes at embryonic stages as evidenced by tendon marker gene expression and by the development of a functional collagen matrix (*Figure 2*). We next sought to determine if the *ScxGFP*-negative cells were indeed tendon cells that lost tendon gene expression or if the mutant tendons were simply repopulated by non-tenogenic cells. Using TUNEL assay, we did not detect cell death in mutant tendons and the rate of tenocyte proliferation as examined by EdU assay was also not altered in these tendons during different developmental stages ranging from E14.5 to P10 (*Figure 2—figure supplement 1A,B*), suggesting the cell population of mutant tendons was not altered. To directly determine if the cells in mutant tendons were tenocytes whose cell fate was altered, we took advantage of the fate mapping feature of the *RosaT* Cre reporter system (*Madisen et al., 2010*). When the reporter is activated by *ScxCre*, expression of the *RosaT* reporter is restricted to the *Scx*-expressing cells and their progeny. We found that all *ScxGFP*-negative cells within mutant tendons were positive for the *RosaT* Cre reporter (*Figure 4D*, white arrowhead). Notably, some non-tendon cells are also positive for the *RosaT* Cre reporter at this

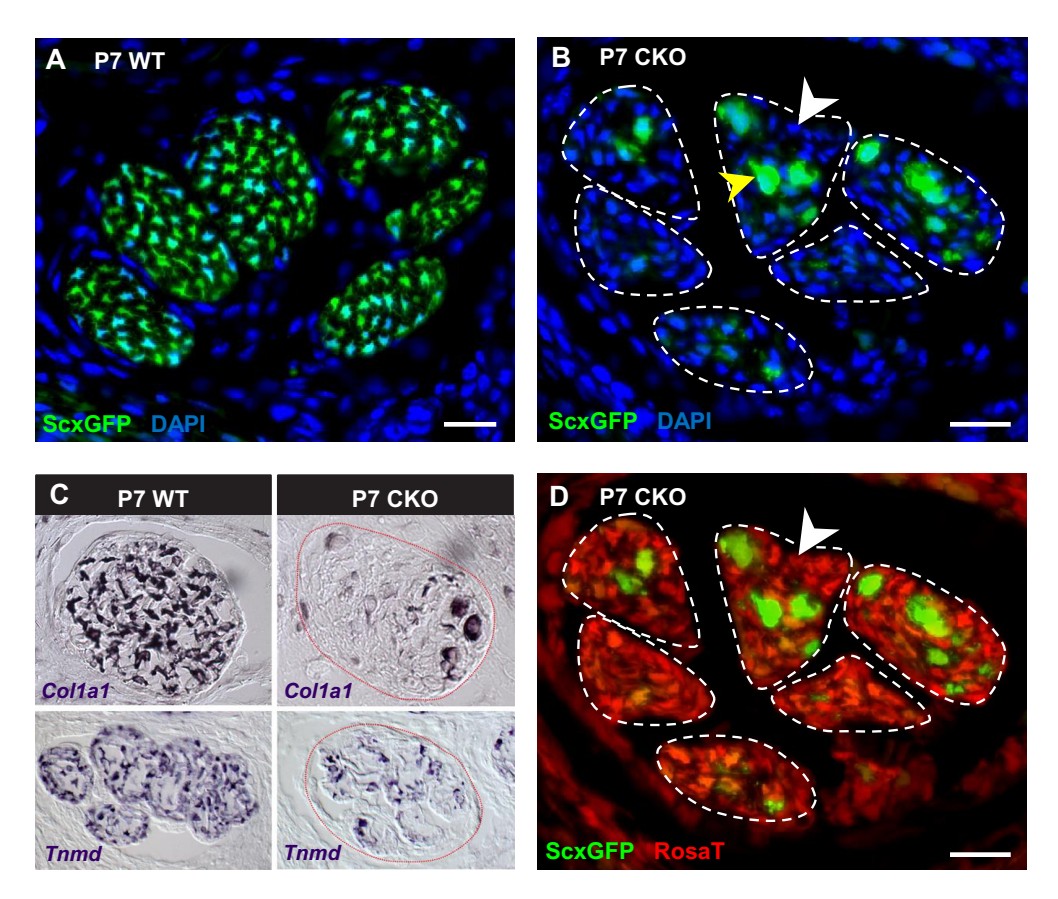

**Figure 4.** Deletion of *Tgfbr2* in *Scx*-expressing cells (*Tgfbr2;ScxCre*) results in loss of tenocyte differentiation markers. (A–D) Transverse sections of extensor digitorium communis tendons of wild-type and mutant pups at wrist level. (A) In P7 wild-type pups, all tenocytes were positive for tendon reporter *ScxGFP* signal. (B) Conversely, most cells in P7 mutant tendons lost the *ScxGFP* signal (white arrowhead), whereas the cells positive for *ScxGFP* signal are newly recruited cells (yellow arrowhead) (Tan et al. in preparation). (C) In situ hybridization shows that the mutant cells also lost gene expression of tendon markers *Col1a1* and *Tnmd* (images not to scale). (D) Lineage tracing using *ScxCre* shows that all *ScxGFP*-negative cells in (B) were positive for *Ai14 Rosa26-tdTomato* (*RosaT*) Cre reporter (white arrowhead), proving that the *ScxGFP*-negative cells in mutant tendons were derived from tenocytes. Dashed lines demarcate the mutant tendons. Scale bar, 20 μm. Mutant: CKO, wild-type: WT.

stage. However, given that there is no apparent elimination of the existing tenocytes, even if some of these cells were recruited into the mutant tendons that would not explain the absence of the original tenocytes in mutant tendons. This result thus indicates that the *ScxGFP*-negative cells in the mutant tendons were derived from tenocytes, and highlighted an unexpected reversibility for the tendon cell fate where it was possible for committed and functional tenocytes to lose their differentiation status.

Next, we wanted to ask if these results reflected that maintenance of the tendon cell fate was dependent on continuous activation of TGFβ signaling. Since the cellular phenotype manifestated mainly in post-natal stage, we targeted *Tgfbr2* in all cells using the ubiquitous tamoxifen-inducible *Rosa*^CreERT2^ driver (*Hameyer et al., 2007*). *Tgfbr2;Rosa*^CreERT2^ pups were fed with tamoxifen at P1, P2 and P5, P6 (1.25 mg per pup for each time point) and harvested at P7-P14. Efficient recombination of the *Tgfbr2* gene was confirmed by immunostaining of the receptor (*Figure 1—figure supplement 1D,E*). Interestingly, the cell fate of targeted cells was not disrupted in these mutants as evidenced by retention of tendon marker expression (*Figure 1—figure supplement 1F*). This result suggests that a mere loss of TGFβ signaling is not sufficient to cause tenocyte dedifferentiation, and additional factors associated with the loss of *Tgfbr2* in the spatial and temporal features determined by *ScxCre* activity may also play a critical role in this process.

## Mutant tenocytes acquired stem/progenitor features

Loss of cell differentiation marker can be the outcome of a few cellular processes, including most notably cell death, change of cell fate (transdifferentiation) or reversion to a less differentiated state (dedifferentiation) (*Cai et al., 2007*; *Talchai et al., 2012*; *Tata et al., 2013*). As aforementioned, we found no apparent cell death in mutant tendons (*Figure 2—figure supplement 1A*). Using histological staining for the prototypic markers of osteocytes, adipocytes and chondrocytes, we found that loss of tendon gene expression in the cells of mutant tendons was also not due to transdifferentiation (*Figure 2—figure supplement 1C*), suggesting that the changes in mutant tendons may reflect a process of cellular dedifferentiation.

One hallmark of cellular dedifferentiation is the loss of differentiation markers, which is what we observed in mutant tendon cells. When cells dedifferentiate they also assume stemness features for example colony-forming potential, and in most cases these cells also acquire expression of stem/progenitor cell markers (*Sun et al., 2011*; *Tata et al., 2013*; *Nusse et al., 2018*). To date, very little is known about the specific gene expression and cellular behavior of embryonic tendon progenitors. The only defined feature of these cells is the expression of the Scx tendon progenitor marker (*Schweitzer et al., 2001*), which was evidently lost in the mutant tendon cells. We therefore next directed our attention to similarities with stem/progenitor cells isolated from tendons (tendon-derived stem/progenitor cells) (*Bi et al., 2007*; *Rui et al., 2010*; *Murchison et al., 2007*; *Mienaltowski et al., 2013*) and with stem/progenitor cell markers reported in other studies (*Blitz et al., 2013*; *Dyment et al., 2013*; *Tan et al., 2013*; *Runesson et al., 2015*; *Yin et al., 2016*).

To test the colony-forming capacity of the mutant tendon cells, P7 mutant tendons were dissociated and FACS-sorted to collect *ScxGFP*-negative and *RosaT*-positive cells, which were then seeded at one cell per well in 96-well plates. As shown in *Figure 5A*, about 1–2% of cells (*ScxGFP*-positive and *RosaT*-positive) isolated from tendons of P7 WT and heterozygous controls formed colonies in

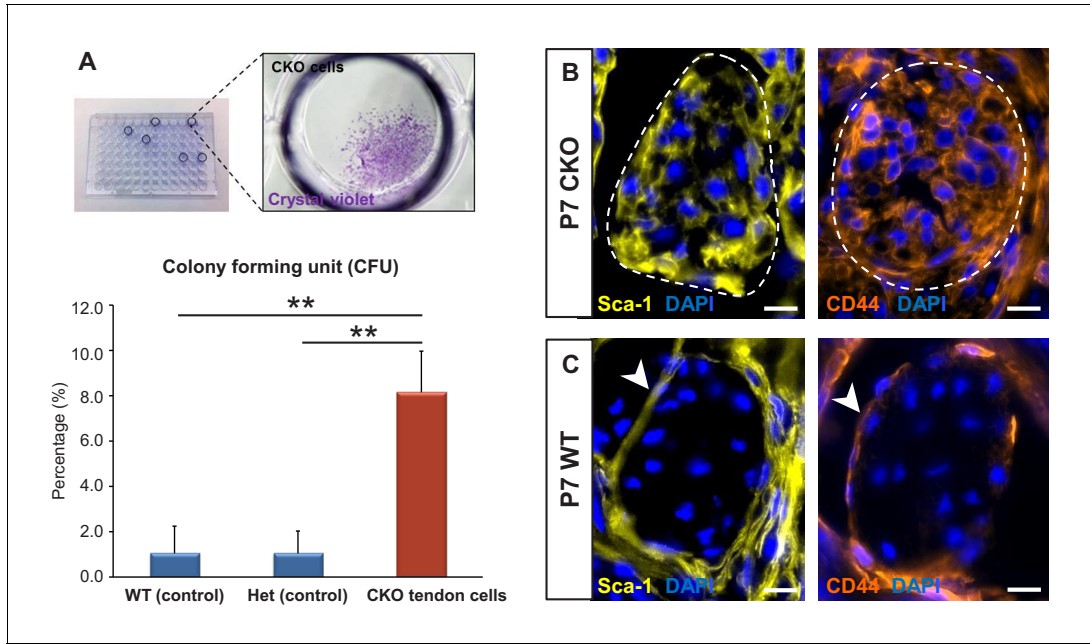

**Figure 5.** *Tgfbr2;ScxCre* mutant tenocytes acquired stem/progenitor features. (**A**) The colony-forming efficiency of P7 wild-type and heterozygous tenocytes as well as mutant tendon cells were evaluated by seeding one cell per well of the FACS-sorted cells in 96-well plates, and colonies formed were visualized with crystal violet staining. Mutant tenocytes exhibited significantly higher clonogenic capacity compared to wild-type and heterozygous controls. The results shown are mean ± SD (n = 5–6, **p<0.01). (**B**) Immunofluorescence staining for stem/progenitor markers in transverse sections of mutant tendons shows that mutant tendon cells acquired in postnatal stages expression of stem cell antigen-1 (Sca-1) and CD44. (**C**) In wild-type littermate controls, expression of both markers was detected in epitenon (white arrowhead), but not in tenocytes. Dashed line demarcates the mutant tendon. Scale bars, 10 μm. Mutant: CKO, Wild-type: WT, Heterozygous: Het.

The online version of this article includes the following figure supplement(s) for figure 5:

**Figure supplement 1.** Expression of Sca-1 and CD44 during embryonic tendon development.

culture, similar to the frequency of colony forming cells reported in other studies (*Bi et al., 2007*; *Rui et al., 2010*). On the other hand, we found a significant eightfold increase (p<0.01) in the frequency of colony-forming cells in mutant tendons (*Figure 5A*).

We next screened the mutant tendons for expression of stem/progenitor cell markers and found that the *Tgfbr2;ScxCre* mutant tendon cells gradually acquired expression of stem cell antigen-1 (Sca-1) and CD44 in postnatal stages (*Figure 5B*). Notably, expression of Sca-1 was undetectable and CD44 was detected only in very few WT tendon cells, but surprisingly robust expression of these markers was detected in the epitenon (*Figure 5C*, white arrowheads), a possible source of progenitor cells (*Mendias et al., 2012*; *Dyment et al., 2013*; *Mienaltowski et al., 2013*; *Harvey et al., 2019*). The similarity of marker expression between the mutant tenocytes and epitenon cells therefore reinforces the notion that the mutant tenocytes acquired progenitor features.

Dedifferentiation is frequently associated with reversion to an earlier progenitor cell fate (*Cai et al., 2007*). We therefore next examined the expression of these markers during embryonic tendon development. At E12.5, when tendon progenitors are first detected (*Pryce et al., 2009*), expression of Sca-1 and CD44 could not be detected in *ScxGFP*-positive tendon progenitors (data not shown). At E14.5, at the onset of tendon cell differentiation, we found low or no expression of both markers in the differentiating tendon cells. Robust positive staining for both markers was however detected in the cells that surround the tendon at this stage, likely the precursors of the epitenon/paratenon (*Figure 5—figure supplement 1*). Similar expression patterns were also found in mutant embryos (data not shown). These findings suggest that Sca-1 and CD44 are not markers for tendon progenitor in vivo, and possibly simply reflect a generic stemness state of the dedifferentiated mutant tendon cells.

Taken together, our findings show that mutant tendon cells acquired some generic stem/progenitor properties while losing their cell fate. It should be noted however that although these dedifferentiated tendon cells demonstrate some stem/progenitor properties, absence of TGFβ signaling in these cells might prevent them from acquiring the full spectrum of stemness or plasticity.

## Molecular profile of the dedifferentiated mutant tenocytes

We next performed single-cell RNA sequencing analysis (scRNASeq) to establish a comprehensive profile of the cellular state and molecular changes in mutant tenocytes. A targeted retention of 2300–2600 cells from P7 WT and mutant tendon was obtained, and the transcriptomes were analyzed using the 10X Genomics platform. Using unsupervised hierarchical clustering analysis, we identified two major clusters corresponding to WT tenocytes and mutant (dedifferentiated) tendon cells in the respective samples (*Figure 6A*, *Supplementary file 1A,B*). The WT tenocyte cluster was defined by the expression of tendon markers including S*cx*, *Fmod*, *Col11a1*, *Col1a1* and *Tnmd*. The mutant (dedifferentiated) tendon cell cluster is enriched for *Ly6a* (encoding Sca-1) and expresses undetectable level of tendon markers. Expression of close to 1000 genes (mean UMI count $\geq 0.5$, adjusted p-value<0.05) was identified in each of these clusters.

Pairwise comparison of the gene sets between the P7 WT tenocyte and mutant cell clusters was next performed to determine changes in gene expression associated with tenocyte dedifferentiation. In total, expression of 186 genes was significantly different between the two cell populations ($\geq 2$ fold change and adjusted p-value<0.05), in which expression of 89 genes was upregulated and 97 genes was downregulated in the mutant tendon cells (*Supplementary file 2*). Almost 30% of the downregulated genes (29 genes) were identified in transcriptome analyses as tendon distinctive genes [(*Havis et al., 2014*) and our unpublished data]. Notably, the genes *Scx, Fmod, Tnmd, Pdgfrl, Col1a1, Col1a2, Col11a1* and *Col11a2* were among the top 25 down-regulated genes in *Tgfbr2; ScxCre* mutant tendon cells (*Table 1*), further confirming the loss-of-cell fate phenotype in these cells. On the other hand, expression of the *Ly6a* gene (encoding Sca-1) was greatly enriched in P7 mutant cells, corroborating the IHC findings presented above (*Table 2* and *Figure 5B*). Moreover, we also found a significant increase in the expression of the *Cd34* gene, another common marker for diverse progenitor cells. This observation was further confirmed at protein level, where positive immunostaining for CD34 was detected in mutant cells but not in normal tendon cells (*Figure 6B*). Interestingly, the genes upregulated in the mutant cells included several genes (*Dpt, Anxa1, Cd34, Cd44, Mgp* and *Mfap5*) whose expression was previously reported to be enriched during embryonic tendon development (*Havis et al., 2014*). These findings thus do not only lend support to our notion that the mutant cells lost their differentiation state, but also suggest the possibility of induction of

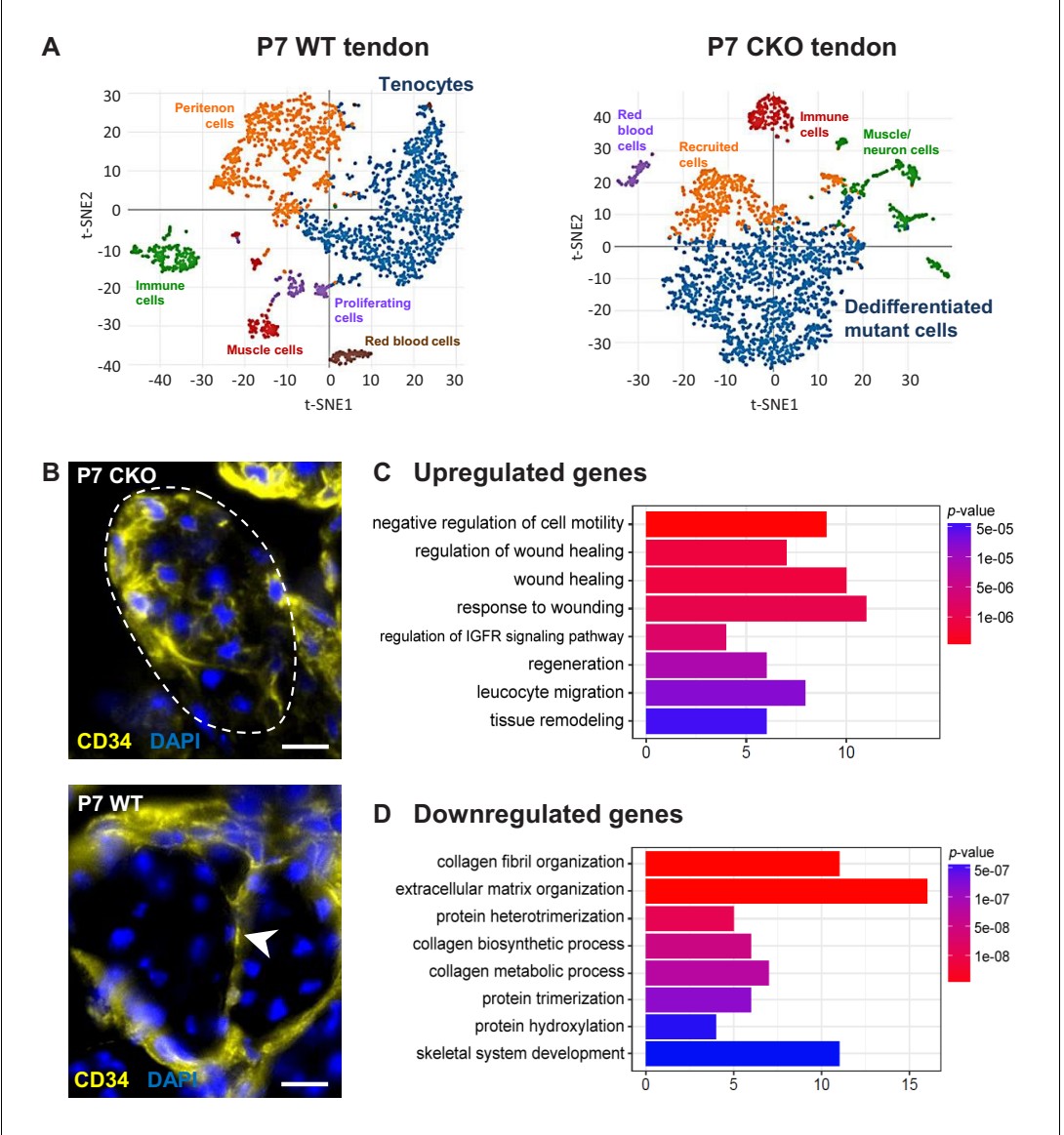

**Figure 6.** Molecular profile of the dedifferentiated mutant tenocytes. (A) tSNE plots (K-means clustering) of enzymatically released cells from P7 wild-type and *Tgfbr2;ScxCre* mutant tendons reveals two major clusters corresponding to tenocytes and dedifferentiated mutant cells in the respective samples. Other cell type assignments are provided in the plots. See ***Supplementary file 1*** for the list of genes highly expressed in these two clusters relative to other clusters. (B) Upregulated expression of *Cd34* gene in P7 mutant tenocytes as revealed by scRNASeq analysis (see also ***Table 2***) was determined using immunostaining. Transverse section of forelimb tendons shows that CD34 was indeed expressed by mutant tenocytes, while in wild-type controls CD34 was detected only in epitenon cells (white arrowhead). Dashed line demarcates the mutant tendon. (C,D) Gene ontology (GO) enrichment analysis in terms of biological processes associated with the (C) upregulated and (D) downregulated genes in P7 mutant compared with wild-type tenocytes. Selected GO terms are included in this figure, and genes annotated to the GO terms are available in ***Supplementary file 3***. Scale bar, 10 µm. Mutant: CKO, Wild-type: WT.

some developmental programs in these cells, a general feature in cellular dedifferentiation (*Tata et al., 2013*; *Stocum, 2017*; *Nusse et al., 2018*).

To gain insights into biological functions activated in the P7 mutant cells, differentially expressed genes (DEGs) in these cells (*Supplementary file 2*) were further analyzed via GO enrichment tools clusterProfiler (*Yu et al., 2012*) and PANTHER Classification System (http://pantherdb.org/). Intriguingly, GO enrichment analysis revealed that one of the prominent biological changes observed in P7 mutant cells was upregulation of gene sets associated with wound healing (*Figure 6C* and *Supplementary file 3A*). These genes include protease inhibitors (*Serpine2, Serping1*), inflammatory

**Table 1.** Top 25 downregulated genes in P7 *Tgfbr2;ScxCre* mutant cells compared with P7 wild-type tenocytes (≥2 fold change, adjusted p<0.05).
See also *Supplementary file 2* for a complete list of the downregulated genes.

| Gene symbol | Gene name | Fold change |
|---|---|---|
| *Wif1* | Wnt inhibitory factor 1 | 157.4 |
| *Col11a2*[#] | Collagen, type XI, alpha 2 | 92.0 |
| *Scx*[#] | Scleraxis | 66.2 |
| *Col2a1*[δ] | Collagen, type II, alpha 1 | 58.9 |
| *Car9* | Carbonic anhydrase 9 | 58.1 |
| *Sema3b* | Sema domain, immunoglobulin domain (Ig), short basic domain, secreted, (semaphorin) 3B | 43.9 |
| *Cgref1* | Cell growth regulator with EF hand domain 1 | 33.2 |
| *Fmod*[#] | Fibromodulin | 27.9 |
| *Cilp2* | Cartilage intermediate layer protein 2 | 24.7 |
| *Matn4* | Matrilin 4 | 19.3 |
| *P4ha1*[δ] | Procollagen-proline, 2-oxoglutarate 4-dioxygenase (proline 4-hydroxylase), alpha one polypeptide | 13.5 |
| *Pcolce2*[δ] | Procollagen C-endopeptidase enhancer 2 | 11.8 |
| *Tpm1* | Tropomyosin 1, alpha | 10.0 |
| *Wisp1* | WNT1 inducible signaling pathway protein 1 | 9.7 |
| *Tnmd*[#] | Tenomodulin | 8.5 |
| *Loxl2*[δ] | Lysyl oxidase-like 2 | 8.3 |
| *1500015O10Rik* | RIKEN cDNA 1500015O10 gene | 7.1 |
| *Col11a1*[#] | Collagen, type XI, alpha 1 | 7.1 |
| *Pdgfrl*[δ] | Platelet-derived growth factor receptor-like | 7.0 |
| *Mfap4* | Microfibrillar-associated protein 4 | 6.5 |
| *Col1a1*[#] | Collagen, type I, alpha 1 | 6.4 |
| *Ptgis* | Prostaglandin I2 (prostacyclin) synthase | 6.4 |
| *Col1a2*[#] | Collagen, type I, alpha 2 | 6.2 |
| *Itgbl1* | Integrin, beta-like 1 | 5.7 |
| *Tpm2* | Tropomyosin 2, beta | 5.4 |

Note:

1) #=Tendon differentiation or specific marker; δ = genes related to tendons.

2) Note that the expression level detected for *Scx* also included that of *ScxGFP*, and therefore do not reflect the expression level of endogenous *Scx*.

mediator *Anxa1* and extracellular matrix (*Col3a1* and *Col5a1*). This finding suggests a possible role for tendon cells in the responses to pathological conditions, in line with findings reported by others (*Dakin et al., 2015*; *Stolk et al., 2017*; *Schoenenberger et al., 2018*). On the other hand, many biological processes downregulated in P7 mutant cells involved collagen synthesis and organization (*Figure 6D* and *Supplementary file 3B*). Since tendon biology is not annotated in most databases, changes in the collagen matrix, the most prominent structural component in tendons is the best indicator for the disruption of the tendon cell fate. Disruption of the collagen matrix in tendons was also detected in older mutant pups by ultrastructural analysis using TEM (*Figure 3E,G*).

Using PANTHER, we also investigated which protein classes were significantly altered in P7 mutant cells relative to WT tenocytes. Genes found to be most downregulated in mutant cells encode for receptors, signaling molecules, membrane traffic proteins and ECM (*Table 3A*). On the other hand, the upregulated genes in the mutant cells encode most prominently for proteins involved in nucleic acid binding, enzyme modulators, cytoskeletal protein, signaling molecules and

transcription factors (*Table 3B*). Notably, expression of the activating protein 1 (AP-1) transcriptional complex, associated with numerous cellular processes including cell fate regulation (*Hess et al., 2004*), was significantly induced in mutant cells. Expression of both AP-1 components, that is the *Fos* and *Junb* genes was induced more than twofold, and the *Jun* gene was induced only slightly less than twofold. Moreover, the *Id3* gene encoding for a general bHLH transcription factor inhibitor was also induced. Due to its broad selection of targets, *Id3* was also implicated in numerous cellular processes including the regulation of cellular differentiation (*Norton, 2000*). A possible role for these transcriptional activities in tenocyte dedifferentiation will be addressed in future studies.

We next conducted PANTHER Pathway Analysis using different values of the filter parameter (mean UMI count and fold change) for enriching DEGs in P7 mutant cells. In general, we found that pathways that stood out as relevant for this study included integrin signaling, insulin/IGF, Wnt and inflammation mediated by chemokine and cytokine signaling pathways (*Table 4*). Insulin/IGF and Wnt signaling are often implicated in cell proliferation and cell fate specification (*Stewart and Rotwein, 1996*; *Sadagurski et al., 2006*; *Goessling et al., 2009*; *Salazar et al., 2016*). It is interesting to note that their activation has also been associated with cellular dedifferentiation in skin, gut and neuron (*Weber et al., 2003*; *Zhang et al., 2012*; *Perekatt et al., 2018*). Further investigation is required to determine the specific roles of these signaling pathways in tenocyte dedifferentiation.

## Tenocyte dedifferentiation is dependent on cell autonomous loss of TGFβ signaling

Lastly we wanted to ask if tenocyte dedifferentiation in these mutants reflected a cell autonomous requirement for TGFβ signaling in tenocytes, or if it was the result of global changes that occurred in mutant tendons. To address this question, we wanted to reactivate TGFβ signaling in isolated mutant tendon cells that will therefore still be exposed to the mutant tendon environment and determine the effects on tenocyte dedifferentiation. We previously found that transuterine injection of AAV viruses into embryonic limbs resulted in sporadic infection of limb tendons [(*Huang et al., 2013*) and unpublished data]. We therefore decided to address this question by injection of a Cre-activatable virus encoding an epitope tagged version of the receptor, *AAV1-FLEX-Tgfbr2-V5* (*Figure 7A*). Injection of this virus into embryonic mutant limbs would result in expression of *Tgfbr2-V5* only in infected tendon cells due to the tendon-restricted activity of *ScxCre* in mutant embryos.

*AAV1-FLEX-Tgfbr2-V5* was injected into mutant limbs at two stages during embryonic tendon development: (a) E12.5 at the onset of *ScxCre* activity, ensuring that *Tgfbr2-V5* expression will be activated in infected cells concurrent with the loss of the endogenous *Tgfbr2*, resulting in isolated *Tgfbr2*-expressing cells surrounded by mutant cells. (b) E16.5, before the onset of tenocyte dedifferentiation in mutant embryos. Infected limbs were harvested at P5-P7, and the effects of *Tgfbr2* expression on mutant tendon cells was evaluated by analyzing cells with positive *V5* signal. Interestingly, targeted re-expression of *Tgfbr2-V5* in individual mutant tendon cell at both developmental stages was able to prevent the loss of tendon markers as observed in postnatal pups (*Figure 7B–D*), suggesting a cell autonomous role for TGFβ signaling in maintenance of the tendon cell fate.

Recognizing that cell autonomous activity of *Tgfbr2-V5* was sufficient to prevent dedifferentiation of mutant tenocytes, we next wanted to test if reactivation of TGFβ signaling in a dedifferentiating tenocyte could also reverse the process and rescue a tenocyte from dedifferentiation. Activity of *ScxCre* may be lost in the dedifferentiating tenocytes due to the loss of *Scx* expression and therefore of *Scx* enhancer driven expression of *Cre* in tendons of *Tgfbr2;ScxCre* mice. We therefore used in this case a virus encoding constitutive expression of *Tgfbr2* in which the virus was tagged with a FLAG Tag (*AAV1-Tgfbr2-FLAG*). The virus was injected locally into P1 mutant limbs and the limbs were harvested at P7. We found again that all infected mutant tendon cells expressed the tendon markers *ScxGFP* and tenomodulin (*Figure 7D* and *Figure 7—figure supplement 1A*), suggesting that reactivation of TGFβ signaling was indeed sufficient to rescue the dedifferentiated tenocytes. Taken together, these findings demonstrate that TGFβ signaling is sufficient to prevent and to rescue the loss-of-tendon cell fate in a cell autonomous manner.

The constitutive expression of *Tgfbr2-FLAG* driven by the *AAV1-Tgfbr2-FLAG* virus ensured that the neonatal infection with this virus resulted in *Tgfbr2-FLAG* expression both within and outside of tendons. Notably, induction of tendon gene expression following activation of *Tgfbr2-FLAG* expression was detected only in dedifferentiated tenocytes and not in cells located outside of tendons (*Figure 7—figure supplement 1B*). It was previously shown that TGFβ signaling is a potent inducer of

**Table 2.** Top 25 upregulated genes in P7 *Tgfbr2;ScxCre* mutant cells compared with P7 wild-type tenocytes (≥2 fold change, adjusted p<0.05).
See also *Supplementary file 2* for a complete list of the downregulated genes.

| Gene symbol | Gene name | Fold change |
| --- | --- | --- |
| Dlk1 | Delta-like one homolog (*Drosophila*) | 137.9 |
| Serpine2 | Serine (or cysteine) peptidase inhibitor, clade E, member 2 | 118.2 |
| Dpt | Dermatopontin | 95.7 |
| Ly6a | Lymphocyte antigen six complex, locus A | 54.3 |
| H19 | H19 | 51.1 |
| Cd34 | CD34 antigen | 47.8 |
| Lum | Lumican | 36.6 |
| Lgmn | Legumain | 31.8 |
| Cxcl12 | Chemokine (C-X-C motif) ligand 12 | 26.1 |
| Mfap5 | Microfibrillar associated protein 5 | 22.5 |
| Ly6c1 | Lymphocyte antigen six complex, locus C1 | 21.7 |
| Igf2 | Insulin-like growth factor 2 | 21.4 |
| Serping1 | Serine (or cysteine) peptidase inhibitor, clade G, member 1 | 19.2 |
| Mgst1 | Microsomal glutathione S-transferase 1 | 18.3 |
| Aspn | Asporin | 15.9 |
| Mt1 | Metallothionein 1 | 15.4 |
| Mgst3 | Microsomal glutathione S-transferase 3 | 13.1 |
| Col3a1[δ] | Collagen, type III, alpha 1 | 13.0 |
| Postn | Periostin, osteoblast specific factor | 13.0 |
| Itm2a | Integral membrane protein 2A | 12.7 |
| Ptn | Pleiotrophin | 10.3 |
| Rps18-ps3 | Ribosomal protein S18, pseudogene 3 | 9.7 |
| Gsn | Gelsolin | 8.3 |
| Ifitm3 | Interferon induced transmembrane protein 3 | 8.2 |
| Col5a1[δ] | Collagen, type V, alpha 1 | 8.1 |

Note: δ = genes related to tendons.

*ScxGFP* and other tendon markers (*Pryce et al., 2009*; *Maeda et al., 2011*; *Sakabe et al., 2018*). This result however, reflects the fact that induction of tendon markers by TGFβ signaling is context-dependent and further indicates that the tenocytes in mutant pups have dedifferentiated to a state that retained tenogenic potential and the capacity to respond to TGFβ signaling.

Taken together these results highlight a surprising cell autonomous role for TGFβ signaling in maintenance of the tendon cell fate. In *Tgfbr2;ScxCre* mutants tenocyte differentiation and function are normal during embryonic development but the tenocytes dedifferentiate in early postnatal stages. Tenocyte dedifferentiation is directly dependent on the loss of TGFβ signaling since retention or reactivation of the TGFβ receptor in isolated cells prevents or reverses the process of dedifferentiation. TGFβ signaling is thus essential for maintenance of the tendon cell fate.

## Discussion

In this study, we find that the tendon cell fate requires continuous maintenance in vivo and identify an essential role for TGFβ signaling in maintenance of the tendon cell fate. To examine the different roles that TGFβ signaling may play in tendon development the *Tgfbr2* gene was targeted in *Scx*-expressing cells (*Tgfbr2;ScxCre* mutant), ensuring disruption of TGFβ signaling in tendon cells. Mutant embryos appeared normal at birth and showed movement difficulties from early neonatal

stages. Tendon formation and maturation was not affected in mutant embryos, but one flexor tendon snapped consistently at E16.5 and a few additional tendons disintegrated in early postnatal stages. Surprisingly, we find that in all other tendons the resident tenocytes lost tendon gene expression and dedifferentiated, assuming behavior and gene expression associated with stem/progenitor cells. While a direct loss of TGFβ signaling in individual tenocytes was not sufficient to cause tenocyte dedifferentiation, we found that tenocyte dedifferentiation could be reversed by reactivation of TGFβ signaling in mutant cells (*Figure 8*). These results uncover an essential role for molecular pathways that maintain the differentiated cell fate in tenocytes and a key role for TGFβ signaling in these processes.

Dedifferentiation has mostly been studied in vitro (*Weinberg et al., 2007*; *Zhang et al., 2010*; *Pennock et al., 2015*; *Mueller et al., 2016*; *Guo et al., 2017*; *Nordmann et al., 2017*) and there are only a handful of reported cases of dedifferentiation in vivo (*Talchai et al., 2012*; *Tata et al., 2013*; *Zhang et al., 2019*). It was therefore important to establish if the tenocytes of *Tgfbr2;ScxCre* mutants indeed dedifferentiated. Cellular dedifferentiation manifests in most cases by loss of features associated with the differentiated state and reversion to an earlier progenitor state within their cell lineage. In tendons of *Tgfbr2;ScxCre* mutants, we indeed found that the tenocytes lost tendon gene expression and showed enhanced clonogenic potential. Moreover, the mutant tenocytes gained expression of the prototypic somatic stem/progenitor markers Sca-1, CD34 and CD44 (*Holmes and Stanford, 2007*; *Sung et al., 2008*; *Hittinger et al., 2013*; *Sidney et al., 2014*). Notably, of these stem/progenitor markers only Sca-1 and CD44 are also expressed at high levels in cultured tendon-derived stem/progenitor cells (*Bi et al., 2007*; *Mienaltowski et al., 2013*). Neither of these markers has so far been established as markers for tenocytes or for tendon progenitors. However, both expression of the *Cd34* and *Cd44* genes and expression of some additional signature genes identified in the dedifferentiated tenocytes by the scRNASeq analysis was previously shown to be significantly enriched in E14.5 mouse limb tendon cells when compared to cells from E11.5 (*Havis et al., 2014*). These observations suggest that some aspects of the embryonic tendon development program may be reactivated in dedifferentiated mutant tendon cells. Interestingly, we found that Sca-1, CD34 and CD44 are expressed in the wild-type epitenon/paratenon, thin layers of cells that surround the tendon and has been implicated as a possible source of stem/progenitor cells for tendons (*Mienaltowski et al., 2013*; *Cadby et al., 2014*; *Harvey et al., 2019*). We further verified that mutant tendons are not repopulated by epitenon/paratenon cells since there is no evidence of elimination of the resident tenocytes by cell death.

Most studies of cellular dedifferentiation have focused on the regulation of this process in vitro. There is, however, evidence demonstrating this phenomenon in vivo especially in the context of pathological scenarios, as part of the regeneration process. One of the well-studied examples is limb regeneration in amphibians. Following limb amputation, cells near to the wound dedifferentiate to blastema, proliferate and eventually re-differentiate to replace all the components of the lost limb (*McCusker et al., 2015*). In zebrafish, it has also been reported that following partial heart amputation, sarcomeres in mature cardiomyocytes disassembled, lost their differentiation gene expression profile and switched to embryonic hyperplastic growth to replace the missing tissues (*Poss et al., 2002*). Cellular dedifferentiation has also been observed in murine mature hepatocytes (*Gournay et al., 2002*), pancreatic β cells (*Talchai et al., 2012*) and skeletal muscle cells (*Mu et al., 2011*). More recently, Nusse and colleagues (*Nusse et al., 2018*) have shown that disruption of the mouse intestinal barrier, via either parasitic infection or cell death, led to reversion of crypt (epithelial) cells to a fetal-like stem cell state. Interestingly, expression of Sca-1 was highly induced in these cells, and when cultured the Sca-1 positive crypt cells exhibited characteristics of fetal intestinal epithelium including re-expression of fetal signature genes and loss of differentiated markers. The results presented in this study therefore suggest that a similar process may be activated in tenocytes as part of the regenerative process in response to pathology. Taken together, this growing body of evidence suggests that dedifferentiation may be a generalized cellular response to tissue damage that warrants further investigation. Moreover, these observations may also suggest that induction of Sca-1 may serve as a marker for a pathology-related dedifferentiation process. Intriguingly, Sca-1-positive cells were also found in the wound window in rat patellar tendon incisional injury model, but in this case it was not determined if Sca-1 expression was associated with dedifferentiation (*Tan et al., 2013*). Sca-1 expression has been identified on putative stem/progenitor cell populations in various tissues (*Holmes and Stanford, 2007*; *Hittinger et al., 2013*), but little is known about its

**Table 3.** PANTHER protein class differentially expressed in P7 *Tgfbr2;ScxCre* mutant cells compared with P7 wild-type tenocytes.

A complete list of differentially expressed genes (≥2 fold change, adjusted p<0.05) used for the analysis is available in ***Supplementary file 2***.

**(A) Downregulated protein class**

| Protein class | Gene list |
| --- | --- |
| Receptor | *Pdgfrl, Col6a3, Kdelr3, Col6a1, Kdelr2, Itgbl1, Ssc5d, Col6a2, Ssr4, Col12a1, Matn4* |
| Signaling molecule | *Sdc1, Wisp1, Sparc, Mfap4, Sema3b, Angptl2, Tgfbi* |
| Membrane traffic protein | *Sec13, Kdelr3, Copz2, Kdelr2, Rabac1, Lman1* |
| Extracellular matrix protein | *Sdc1, Crtap, Clec11a, P3h3, Sparc, P3h4* |

**(B) Upregulated protein class**

| Protein class | Gene list |
| --- | --- |
| Nuclei acid binding | *Ndn, Eif3f, Rpl39, Rpl36a, Rpl3, Rpl9-ps6, Rpl22l1, Rps27, Rps4x, Cirbp, Rps19, Eif3e, Rps18, Rps5, Junb* |
| Enzyme modulator | *Fstl1, Dbi, Sfrp2, Ctsb, Serpine2, Serping1, Igfbp3, Igfbp4* |
| Cytoskeletal protein | *Gsn, Map1lc3b, Tuba1b, Arpc1b, Emp1, Tubb5* |
| Signaling molecule | *S100a16, Ptn, Dlk1, Efemp2, Postn, Sfrp2* |
| Transcription factor | *Eif3h, Naca, Fos, Id3, Junb* |

biological function. It may therefore be interesting to examine whether Sca-1 functions as a stemness marker in dedifferentiated cells or if it also plays additional roles in cellular responses to pathological conditions.

Tenocyte dedifferentiation as observed in this study reveals an unexpected flexibility in the tendon cell fate where differentiated tenocytes can revert to a progenitor state under the mutant conditions. Significantly, reintroduction of *Tgfbr2* not only prevented tenocyte dedifferentiation when it was performed during embryogenesis but was also able to rescue the cell fate of dedifferentiated tenocytes when the virus was introduced after birth. This result suggests that TGFβ signaling may have a continuous role in protecting the differentiated tenocytes from dedifferentiation, identifying TGFβ signaling as a key regulator of tendon homeostasis. Moreover, these results also highlight the importance of the molecular pathways involved in maintenance of the differentiated cell fate not only for tissue homeostasis and function, but also for processes associated with tissue regeneration or with the onset and unfolding of pathology. Previous studies have implicated TGFβ signaling in cell fate maintenance in various tissues, for example preserving chondrocyte identity in cultures (*Baugé et al., 2013*) and suppressing intestinal cell dedifferentiation (*Cammareri et al., 2017*).

**Table 4.** PANTHER pathway analysis of upregulated genes in P7 *Tgfbr2;ScxCre* mutant cells compared with P7 wild-type tenocytes.

| PANTHER pathway | PANTHER accession | Gene list |
| --- | --- | --- |
| Integrin signaling pathway | P00034 | *Arpc2, Col4a1, Rac1, Col5a2, Rap1b, Cdc42, Arpc5, Col5a1, Rap1a, Rhoc, Fn1, Arpc1b, Col3a1* |
| Inflammation mediated by chemokine and cytokine signaling pathway | P00031 | *Arpc2, Rac1, Cdc42, Nfkbia, Arpc5, Rhoc, Arpc1b, Arpc4, Jun, Junb* |
| Wnt signaling pathway | P00057 | *Fstl1, Sfrp2, Ppp3ca, Csnk1a1* |
| Insulin/IGF pathway | P00032, P00033 | *Igf1, Igf2, Fos* |

Note:

[1]A complete list of differentially expressed genes (DEGs) used for the analysis is available in ***Supplementary file 2***.

[2]Different values of the filter parameter (mean UMI count and fold change) were applied for enriching DEGs in P7 mutant cells. Only pathways that stood out as relevant for this study are listed.

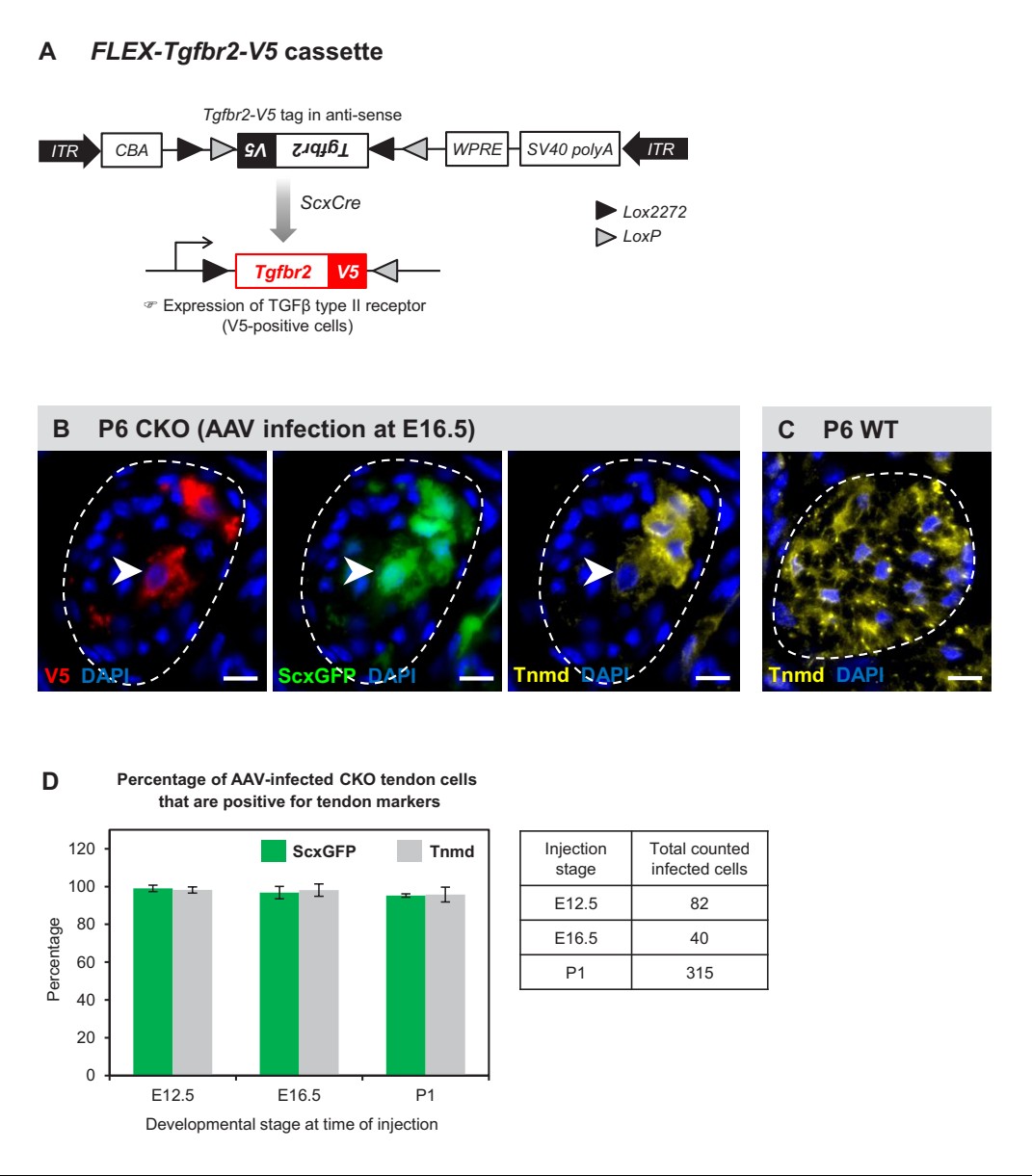

**Figure 7.** Tenocyte dedifferentiation is dependent on cell autonomous loss of TGFβ signaling. (**A**) *AAV1-FLEX-Tgfbr2-V5* virus contains the reverse-complement sequence of *Tgfbr2* with a C-terminal *V5* epitope tag. *Cre* activity will lead to a permanent inversion of the cassette that will then express the V5-tagged TGFβ type II receptor. (**B**) Targeted expression of TGFβ type II receptor in E16.5 mutant tendon cells using the *AAV1-FLEX-Tgfbr2-V5* prevented the loss of tendon markers in the infected tenocytes. The forelimb of E16.5 mutant embryos was infected with *AAV1-FLEX-Tgfbr2-V5* virus and harvested at P6. Transverse forelimb sections were stained with antibodies for V5 (red) to detect AAV-infected cells and tenomodulin (Tnmd; yellow), a prototypic tendon marker expressed by (**C**) all tenocytes in the wild-type tendon at this stage. Dashed line demarcates the mutant tendon. (**D**) Quantification shows that about 95–98% of the AAV-infected (V5-positive) mutant tendon cells retained or re-expressed tendon differentiation markers after viral injection at different developmental stages (n = 3 pups for each stage). Note that the embryonic infection was performed with Cre-activated *AAV1-FLEX-Tgfbr2-V5* virus and the P1 infection was performed with the constitutive *AAV1-Tgfbr2-FLAG* virus. Scale bar, 10 μm. Mutant: CKO, Wild-type: WT.

The online version of this article includes the following figure supplement(s) for figure 7:

**Figure supplement 1.** Induction of tendon markers by TGFβ signaling is context dependent.

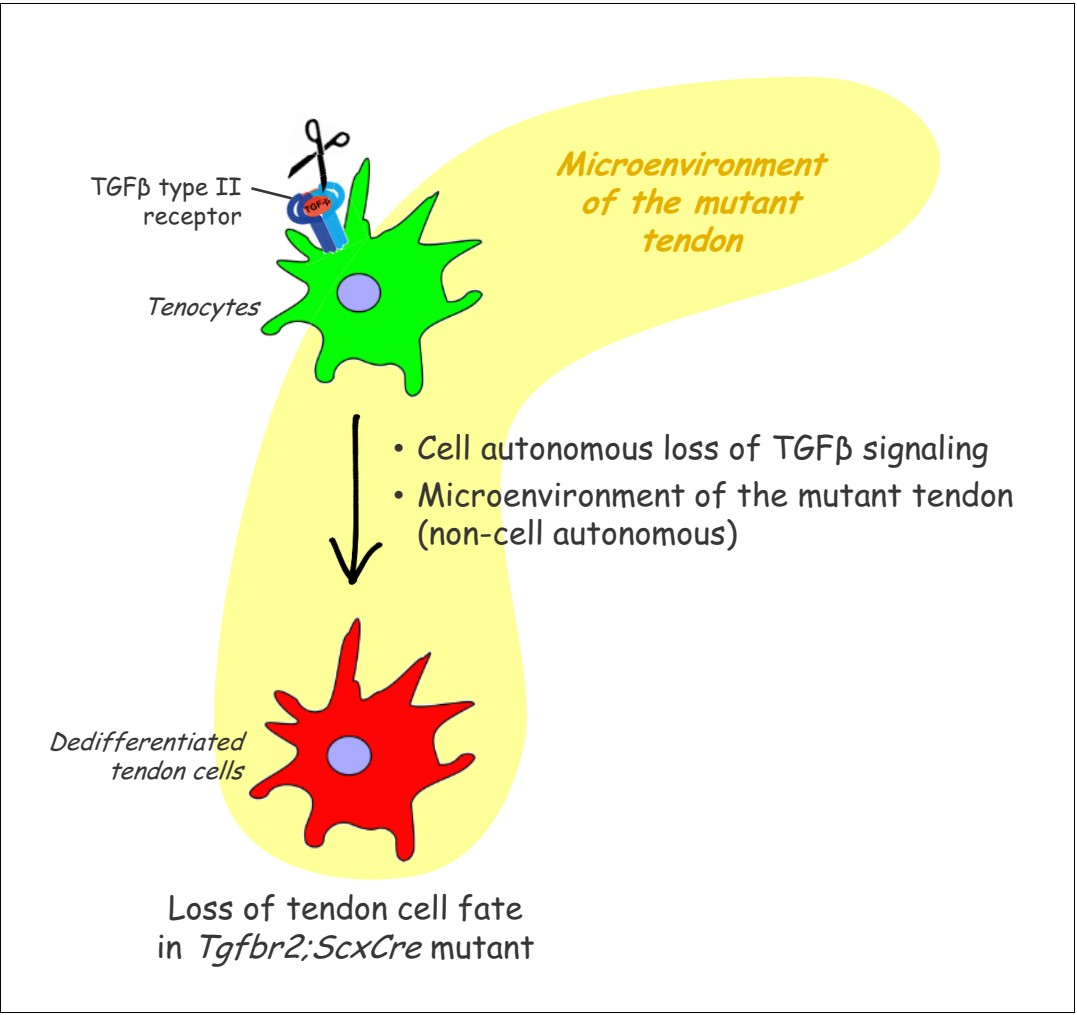

**Figure 8.** Proposed roles of TGFβ signaling in the maintenance of tendon cell fate. Targeted disruption of the TGFβ type II receptor (*Tgfbr2*) by *ScxCre* resulted in tenocyte dedifferentiation in early postnatal stages. Tenocyte dedifferentiation was reversed by reactivation of TGFβ signaling in individual mutant cells, demonstrating a cell autonomous role for TGFβ signaling for maintenance of the cell fate. Conversely, a mere loss of the receptor in individual tendon cell was not sufficient to cause tenocyte dedifferentiation, suggesting that external factors may also play a critical role in this process. We therefore propose that maintenance of the tendon cell fate is dependent on a combination of a cell autonomous function of TGFβ signaling and an additional, likely non-cell autonomous factor, for example the microenvironment of the tendon in the *Tgfbr2;ScxCre* mutant (cell-matrix interaction, mechanical loading, cell-cell contacts etc).

While TGFβ signaling has been associated with different aspects of tendon biology (*Pryce et al., 2009*; *Havis et al., 2016*), to the best of our knowledge this is the first report of its role in maintenance of the tendon cell fate.

The fact that the mutant phenotype was caused by disruption of TGFβ signaling in tenocytes and the rescue of the tendon cell fate by virus mediated reintroduction of *Tgfbr2* even to individual mutant cells provides direct evidence for a continuous and cell autonomous role for TGFβ signaling in maintenance of the tendon cell fate. However, targeting of *Tgfbr2* using ubiquitous inducible cre drivers did not result in tenocyte dedifferentiation. These observations suggest that tenocyte dedifferentiation in these mutants may not merely be the result of the loss of intrinsic TGFβ signaling in tendon cells, but rather may be caused by an interplay between intrinsic loss of TGFβ signaling and additional external factors associated with the loss of *Tgfbr2* with the specific spatial and temporal features of the *ScxCre* driver. These additional factors may involve cell-matrix interactions affected by the microenvironment of the mutant tendons or changes in cell-cell contacts in the mutant

environment. The fact that the phenotype manifested in early post-natal stages may also suggest that mechanical loading experienced by the pups after birth may play a role in the initiation of cellular dedifferentiation. The close relationship between tendon function and mechanical stimulus has been underlined in several studies (*Nabeshima et al., 1996*; *Heinemeier and Kjaer, 2011*; *Galloway et al., 2013*). Enhanced mechanical loading may compound with altered features in the structure of the mutant tendons to trigger the initiation of the mutant phenotype.

The tendon phenotype of *Tgfbr2;ScxCre* mutants highlights a likely role for tenocyte dedifferentiation in regenerative processes in tendons and possibly also in the progression of tendon pathology. Uncovering the molecular pathways involved in this process may therefore be important for new strategies for treatments of tendon pathologies. The *Tgfbr2;ScxCre* mutants provide a unique opportunity to analyze these pathways, and the experimental approaches employed in this study may be developed into an experimental paradigm for molecular dissection of this process. Briefly, transcriptional and epigenetic analyses of the mutant tenocytes through the dedifferentiation process can provide a landscape of the molecular changes that initiate and drive the dedifferentiation process. Promising candidates can then be tested using the AAV-mediated cell fate rescue experiments to identify genes or groups of genes that can protect the tenocytes from dedifferentiation to establish the molecular process of cellular dedifferentiation. Of particular interest will be the early molecular changes in the mutant tenocytes that drive and promote the onset and progression of tenocyte dedifferentiation.

Our findings underscore the fact that the tendon cell fate requires continuous maintenance and that it is not an irreversible state, a long-standing biological dogma that has been challenged by recent research (*Takahashi and Yamanaka, 2006*; *Ladewig et al., 2013*). Nevertheless, it is important to recognize that the dramatic cell fate changes in *Tgfbr2;ScxCre* mutant happens in the context of a genetic modification. The occurrence of such phenomenon in vivo might not be a simple direct outcome of changes to TGFβ signaling. Most importantly, while the initiating events for tenocyte dedifferentiation may vary in different scenarios, it is likely that the molecular events that drive the dedifferentiation process downstream of the initiation event are similar or related. Uncovering these pathways in this experimental system may therefore facilitate the analysis of such processes in various other contexts.

# Materials and methods

**Key resources table**

| Reagent type (species) or resource | Designation | Source or reference | Identifiers | Additional information |
|---|---|---|---|---|
| Genetic reagent (M. musculus) | Tgfbr2^{f/f} | (*Chytil et al., 2002*) | NA | NA |
| Genetic reagent (M. musculus) | ScxCre | (*Blitz et al., 2013*) | NA | NA |
| Genetic reagent (M. musculus) | Rosa^{CreERT} | (*Hameyer et al., 2007*) | NA | NA |
| Genetic reagent (M. musculus) | ScxGFP | (*Pryce et al., 2007*) | NA | NA |
| Genetic reagent (M. musculus) | Ai14 Rosa26-tdTomato (RosaT) | (*Madisen et al., 2010*) | NA | NA |
| Recombinant DNA reagent | pAAV1-FLEX-Tgfbr2-V5 | GenScript | This paper | NA |
| Recombinant DNA reagent | pAAV1-Tgfbr2-FLAG | GenScript | This paper | NA |
| Antibody | Rat anti-CD34 (Clone RAM34) | BD Biosciences | Cat# 553731 RRID:AB_395015 | IF(1:200), Antigen retrieval |
| Antibody | Rat anti-CD44 (Clone IM7) | BD Biosciences | Cat# 550538 RRID:AB_393732 | IF(1:40), Pre-treated with cold acetone for 10 min at −20°C |

*Continued on next page*

*Continued*

| Reagent type (species) or resource | Designation | Source or reference | Identifiers | Additional information |
|---|---|---|---|---|
| Antibody | Rabbit anti-FLAG (DYKDDDDK) | Thermo Fisher Scientific | Cat# 740001 RRID:AB_2610628 | IF(1:200), Antigen retrieval |
| Antibody | Rat anti-FLAG (DYKDDDDK) | Novus Biologicals | Cat# NBP1-06712SS RRID:AB_1625982 | IF(1:100), Antigen retrieval |
| Antibody | Goat anti-Sca-1/Ly6 | R and D Systems | Cat# AF1226 RRID:AB_354679 | IF(1:80) |
| Antibody | Rat anti-Sca-1/Ly6 | R and D Systems | Cat# MAB1226 RRID:AB_2243980 | IF(1:50) |
| Antibody | Goat anti-tenomodulin (Clone C-20) | Santa Cruz Biotechnology | Cat# sc-49324 RRID:AB_2205971 | IF(1:50), Antigen retrieval |
| Antibody | Rabbit anti-TGFβ type II receptor | Bioworld Inc | Cat# BS1360 RRID:AB_1663474 | IF(1:250) |
| Antibody | Rabbit anti-V5 | Abcam | Cat# ab206566 RRID:AB_2819156 | IF(1:500), Antigen retrieval |
| Antibody | Rat anti-V5 | Abcam | Cat# ab206570 RRID:AB_2819157 | IF(1:500), Antigen retrieval |
| Antibody | Cy5 donkey anti-goat secondary | Jackson ImmunoResearch | Cat# 705-175-147 RRID:AB_2340415 | IF(1:500) |
| Antibody | AlexaFluor647 donkey anti-rabbit secondary | Jackson ImmunoResearch | Cat# 711-607-003 RRID:AB_2340626 | IF(1:400) |
| Antibody | Cy3 donkey anti-rabbit secondary | Jackson ImmunoResearch | Cat# 711-166-152 RRID:AB_2313568 | IF(1:800) |
| Antibody | AlexaFluor647 donkey anti-rat secondary | Jackson ImmunoResearch | Cat# 712-606-153 RRID:AB_2340696 | IF(1:800) |
| Antibody | Cy3 donkey anti-rat secondary | Jackson ImmunoResearch | Cat# 712-166-150 RRID:AB_2340668 | IG(1:800) |
| Commercial assay or kit | In situ cell death detection kit | Roche | Cat# 12156792910 | Follow the manufacturer's instruction |
| Commercial assay or kit | Click-iT EdU kit | Life Technologies | Cat# C10340 | Follow the manufacturer's instruction |
| Other | DAPI stain | Thermo Fisher Scientific | D1306 RRID:AB_2629482 | 1 µg/ml |

Note:

* Antigen retrieval: Incubated with warm citrate buffer (10 mM sodium citrate with 0.05% Tween 20, pH 6) at 550W, 50°C for 5 min using a PELCO BioWave.

## Mice

All mouse works were performed in accordance to the guidelines issued by the Animal Care and Use Committee at Oregon Health and Science University (OHSU). Floxed TGFβ type II receptor (*Tgfbr2^{f/f}*) mice (**Chytil et al., 2002**) were crossed with mice carrying the tendon deletor Scleraxis-Cre recombinase (*ScxCre*) (**Blitz et al., 2013**) to disrupt TGFβ signaling in tenocytes (called hereafter *Tgfbr2; ScxCre* mutant). All mice in this study also carried a transgenic tendon reporter *ScxGFP* (**Pryce et al., 2007**), and a Cre reporter *Ai14 Rosa26-tdTomato* (*RosaT*) (**Madisen et al., 2010**) for the lineage tracing of *Scx*-expressing cells. For embryo harvest, timed mating was set up in the afternoon, and identification of a mucosal plug on the next morning was considered 0.5 days of gestation (E0.5). Embryonic day 14.5 to postnatal day 13 (E14.5-P13) limb tendons were used for analysis. Mouse genotype was determined by PCR analysis of DNA extracted from tail snip using a lysis reagent (Viagen Biotech, Cat 102 T) and proteinase K digestion (55°C, overnight).

## Transmission electron microscopy (TEM)

Skinned mouse forelimbs were fixed intact for several days in 1.5% glutaraldehyde/1.5% formaldehyde, rinsed, then decalcified in 0.2 M EDTA with 50 mM TRIS in a microwave (Ted Pella, Inc) operated at 97.5 watts for fifteen 99 min cycles. Samples were fixed again in 1.5% glutaraldehyde/1.5%

formaldehyde with 0.05% tannic acid overnight, then rinsed and post-fixed overnight in 1% OsO$_4$. Samples were dehydrated and extensively infiltrated in Spurr's epoxy and polymerized at 70℃ (*Keene and Tufa, 2018*). Ultrathin sections of tendons of interest were cut at 80 nm, contrasted with uranyl acetate and lead citrate, and imaged using a FEI G20 TEM operated at 120 kV with montages collected using a AMT XR-41 2 × 2K camera. The acquired images were stitched using ImageJ software (https://imagej.nih.gov/ij/) (*Preibisch et al., 2009*). Three pups per time point were harvested for TEM analysis.

## In situ hybridization and histological staining

Dissected forelimbs were fixed with 4% paraformaldehyde in PBS, decalcified in 5 mM EDTA (1–2 weeks at 4℃) and incubated with a 5–30% sucrose/PBS gradient. The tissues were then embedded in OCT (Tissue-Tek, Inc), sectioned at 10 μm or 12 μm using a Microm HM550 cryostat (Thermo Scientific, Waltham, MA) and mounted on Superfrost plus slides (Fisher). In situ hybridization was performed as previously described (*Murchison et al., 2007*).

For immunofluorescence staining, sections were air-dried, rinsed thrice with PBS and blocked with 2% BSA and 2% normal goat serum in PBS for 1 hr at RT. The sections were then incubated overnight at 4℃ with specific primary antibody as listed in Key Resources Table. This was followed by incubation with the matching Cy3- or Cy5/AlexaFluor647-conjugated secondary antibody (Jackson ImmunoResearch; diluted at 1:400 to 1:800; see Key Resources Table) in PBS containing 2% normal goat serum for 1 hr at RT. DAPI (4′,6-diamidino-2-phenylindole, dihydrochloride; Thermo Fisher Scientific) was used to counterstain cell nuclei. Immunolabelled sections were mounted in Fluorogel (Electron Microscopy Sciences, PA; Cat 17985–10) and visualized using a Zeiss ApoTome microscope. A washing step with PBS containing 0.1% Triton-X 100 was performed after the change of antibodies. Controls included omission of primary antibodies.

For examination of cell death and proliferation, TUNEL and EdU assays were performed using Click-iT EdU (Life Technologies) and In Situ Cell Death Detection (Roche) kits, respectively, following manufacturer's instructions. For all studies, sections from two to four pups were examined to ensure reproducibility of results.

## Isolation and culture of tendon-derived stem/progenitor cells

Mice at P7 were used for tendon progenitor cell isolation using a protocol modified from that in *Mienaltowski et al. (2013)*. Briefly, both forelimbs and hindlimbs were harvested from euthanized mice, skinned and exposed to 0.5% collagenase type I (Gibco, Cat 17100–017) and 0.25% trypsin (Gibco, Cat 27250–018) in PBS for 15 min at 37℃ with gentle shaking. The surfaces of tendons were then scraped carefully with a pair of forceps to remove epitenon/paratenon cells. The middle portion of tendons was then harvested, cut into small pieces and tendon cells were released by digestion for 30 min at 37℃ with gentle shaking in a solution of 0.3% collagenase type I, 0.8% collagenase type II (Cat 17101–015), 0.5% trypsin and 0.4% dispase II (Cat 17105–041) (all from Gibco). The released cells were strained with a 70 μm cell strainer (BD Falcon, Cat 352350) and collected by centrifugation for 5 min at 300 g. The cells were then resuspended in PBS with 1% BSA, and fluorescence-activated cell sorting (FACS) was used to separate the cells for colony-forming assay.

## Colony-forming unit (CFU) assay

CFU assay was used to examine the self-renewal potential of cells (*Bi et al., 2007*). The enzymatically-released WT and heterozygous tenocytes as well as dedifferentiated mutant tendon cells (i.e. *ScxGFP*-negative and RosaT-positive cells) were sorted by FACS and plated at one cell per well in a 96-well plate using a BD Influx cell sorter (BD Bioscience, USA). About 10–12 days into the culture, the colonies were fixed in 4% paraformaldehyde (10 min, RT), stained with 0.5% crystal violet for 30 min, and rinsed twice with water. Percentage of colony-forming unit was calculated as: Number of wells with colonies ÷ 96 × 100. Each data point represents the mean of duplicate plates from 3 to 5 separate experiments. Each experiment represents limb tendons collected from 2 to 4 pups.

## Re-expression of *Tgfbr2* in mutant cells using adeno-associated virus (AAV) vector

*FLAG* or *V5* epitope tag sequences were added at the C-terminus of the murine TGFbR2 Consensus Coding Sequence (CCDS23601). The *Tgfbr2-FLAG* (*Tgfbr2-FLAG*) and reverse-compliment *Tgfbr2-V5* (*FLEX-Tgfbr2-V5*) insert sequences were synthesized and subcloned by GenScript into an *AAV1* vector. The *FLEX* backbone vector (*Atasoy et al., 2008*) was purchased from AddGene and modified. Vectors were then packaged into *AAV1* capsid, purified, and titered by the OHSU Molecular Virology Support Core. *AAV1* insert expression was under the control of a chicken beta-actin (*CBA*) promoter and an *SV40* polyadenylation sequence. All experimental procedures were evaluated and approved by the institutional Animal Care and Ethics Committee.

Re-expression of *Tgfbr2* in embryos was done by delivery of *AAV1-FLEX-Tgfbr2-V5*, a Cre-dependent expression cassette, specifically to *Tgfbr2;ScxCre* mutant tendon cells. Transuterine microinjection of the viral vector into embryos was performed according to a published protocol (*Jiang et al., 2013*). Briefly, a laparotomy was performed on anesthetized pregnant females to expose the uterus. The left wrist field of the forelimb bud of each embryo was injected with ~2 μl of concentrated viral inoculum ($3.8 \times 10^{13}$ vg/ml) using a borosilicate glass capillary pipette (25–30 μm outer diameter and 20 degree bevel). The abdominal and skin incisions were closed with resorbable sutures. The dams were recovered overnight with supplementary heating and then returned to main colony housing.

For postnatal constitutive re-expression of *Tgfbr2*, ~10 μl of *AAV1-Tgfbr2-FLAG* inoculum ($4.1 \times 10^{12}$ vg/ml) was injected subcutaneously into the left forelimb of P1 pups using an 8 mm x 31G BD Ultra-Fine insulin syringe and needle (Becton Dickinson and Company, NJ). For both experiments, forelimbs from P5 to P7 mutant pups (n = 3 pups for each stage) were harvested, fixed, cryosectioned and examined for expression of tendon differentiation markers in infected tendon cells.

## Single-cell RNA sequencing (scRNASeq) and data analysis

Tendons were collected and pooled from both forelimbs and hindlimbs as described above from two pups with the omission of tissue-scraping step. The enzymatically released cells were centrifuged, resuspended in α-MEM with 5% FBS and submitted to the OHSU Massively Parallel Sequencing Shared Resource (MPSSR) Core facility. scRNASeq analysis was then performed using the 10x Genomics Chromium Single Cell 3′ Reagent Kits and run on a Chromium Controller followed by sequencing using the Illumina NextSeq 500 Sequencing System (Mid Output), as per the manufacturer's instructions (10x Genomics Inc, CA; Illumina Inc, CA).

Sequencing data processing and downstream analysis were performed using Cell Ranger version 2.0 (10x Genomics, CA) (*Zheng et al., 2017*) with the default settings. Briefly, sequencing reads were aligned to the mm10 genome and demultiplexed and filtered using total UMI count per cell to generate the gene barcode matrix. Principle component analysis was performed and the first ten principle components were used for the t-distributed stochastic neighbor embedding (tSNE) dimensional reduction and clustering analysis. Cells were clustered using K-means clustering. For each cluster, genes with an average UMI count $\geq 0.5$, fold change $\geq 1.5$ and p-value<0.05 were identified as signature genes for each cluster. Gene Ontology (GO) enrichment analysis (clusterProfiler) (*Yu et al., 2012*) and the PANTHER Classification System (http://pantherdb.org/) were used to elucidate the biological process and signaling pathway associated with individual gene. Enriched canonical pathways were defined as significant if adjusted *p*-values were <0.05.

## Statistical analysis

Unless stated otherwise, all graphs are presented as mean ± standard deviation (SD). Student's *t*-tests were performed to determine the statistical significance of differences between groups (n ≥ 3). A value of p<0.05 is regarded as statistically significant.

## Acknowledgements

The authors thank Dr Elazar Zelzer (Department of Molecular Genetics, Weizmann Institute of Science, Israel) for critical reading of the manuscript. We are grateful to staff from MPSSR and FACS core facilities, OHSU particularly Dr Robert Searles, Mrs Amy Carlos and Dr Miranda Gilchrist for

their excellent technical assistance. This work was funded by NIH (R01AR055973, RS; R01DC014160, JVB) and Shriners Hospitals for Children (SHC 5410-POR-14). G.K.T was supported by Research Fellowship from Shriners Hospital for Children.

## Additional information

### Funding

| Funder | Grant reference number | Author |
|---|---|---|
| National Institutes of Health | R01AR055973 | Ronen Schweitzer |
| Shriners Hospitals for Children | SHC 5410-POR-14 | Ronen Schweitzer |
| National Institutes of Health | R01DC014160 | John V Brigande |

The funders had no role in study design, data collection and interpretation, or the decision to submit the work for publication.

### Author contributions

Guak-Kim Tan, Conceptualization, Data curation, Formal analysis, Validation, Investigation, Visualization, Methodology, Project administration; Brian A Pryce, Anna Stabio, Resources, Data curation, Investigation; John V Brigande, Funding acquisition, Investigation; ChaoJie Wang, Zheng Xia, Software, Formal analysis; Sara F Tufa, Douglas R Keene, Investigation; Ronen Schweitzer, Conceptualization, Supervision, Funding acquisition, Project administration

### Author ORCIDs

Ronen Schweitzer (iD) https://orcid.org/0000-0002-7425-5028

### Ethics

Animal experimentation: This study was performed in strict accordance with the recommendations in the Guide for the Care and Use of Laboratory Animals of the National Institutes of Health. All of the animals were handled according to approved institutional animal care and use committee (IACUC) protocols IP00000717 and IP00000935 of the Oregon Health & Science University.

### Decision letter and Author response

Decision letter https://doi.org/10.7554/eLife.52695.sa1
Author response https://doi.org/10.7554/eLife.52695.sa2

## Additional files

### Supplementary files

• Supplementary file 1. Signature genes in tenocytes and dedifferentiated mutant cells in comparison with other clusters. See also *Figure 6A* for the tSNE plots of the sample. (**A**) Top 25 genes highly expressed in the tenocyte cluster relative to other clusters in the P7 wild-type tendon sample ($\geq$1.5 fold change, adjusted p<0.05). (**B**) Top 25 genes highly expressed in the dedifferentiated mutant cell cluster relative to other clusters in the P7 *Tgfbr2;ScxCre* mutant tendon sample ($\geq$1.5 fold change, adjusted p<0.05).

• Supplementary file 2. Differentially expressed genes in P7 *Tgfbr2;ScxCre* mutant tendon cells compared with P7 wild-type tenocytes ($\geq$2 fold change, adjusted p<0.05). Note that the expression level detected for *Scx* also included that of *ScxGFP*, and therefore do not reflect the expression level of endogenous *Scx*.

• Supplementary file 3. Gene Ontology (GO) term enrichment of differentially expressed genes in P7 *Tgfbr2;ScxCre* mutant cells compared with P7 wild-type tenocytes. A complete list of differentially expressed genes ($\geq$2 fold change, p<0.05) used for the analysis is available in *Supplementary file 2*.

• Transparent reporting form

## Data availability

All data generated or analyzed during this study are included in the manuscript and Supplementary Files. Single cell RNA-Seq data has been deposited onto GEO under accession code GSE139558.

The following dataset was generated:

| Author(s) | Year | Dataset title | Dataset URL | Database and Identifier |
|---|---|---|---|---|
| Tan G, Wang C, Xia Z, Schweitzer R | 2020 | Differentially expressed transcriptomes of P7 mouse tendon cells with targeted deletion of TGF-beta signaling | http://www.ncbi.nlm.nih.gov/geo/query/acc.cgi?acc=GSE139558 | NCBI Gene Expression Omnibus, GSE139558 |

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
