## [Decision Letter]

**Acceptance summary:**

Overall, all the reviewers of this study found it to be highly thought provoking and definitely ground-breaking. That this paper convincingly challenged the canonical role of TGFβ in the tendon generated considerable positive discussion during the review of this manuscript. Many researchers in the field believe that TGFβ is critical to tendon growth/maturation via a role in the regulation of collagen synthesis and matrix assembly. This study suggests that TGFβ's role is more so related to the maintenance of tenogenic cell fate. Further, by demonstrating the temporal aspects of the phenotypic changes (i.e., loss of tenogenic cell fate prior to significant matrix changes), this paper shows that dedifferentiation of the tenocyte to a general stem/progenitor state is possible. Collectively this work has high potential to have considerable impact in shaping research in the tendon field for years to come.

**Decision letter after peer review:**

Thank you for submitting your article "TGF-β signaling is critical for maintenance of the tendon cell fate" for consideration by *eLife*. Your article has been reviewed by Clifford Rosen as the Senior Editor, a Reviewing Editor, and three reviewers. The following individuals involved in review of your submission have agreed to reveal their identity: Nathanial Dyment (Reviewer #1); Alayna E Loiselle (Reviewer #2).

The reviewers have discussed the reviews with one another and the Reviewing Editor has drafted this decision to help you prepare a revised submission.

Summary:

This study continues years of work by this lab studying the role TGFβ signaling in tendon differentiation. Previously this group demonstrated that TGFβ is required for tendon specification and formation (Pryce, 2009). In the current study, they used ScxCre mice to target tendon cells after specification and established that TGFβ signaling needs to be maintained by tendon cells to continue their differentiation and preserve a normal ECM. Interestingly, the cell phenotype and ECM does not appear to be abnormal until approximately 7 days after birth. After this stage, the tendons begin to disintegrate and the mutant mice actually die before P14. Analysis of single-cell RNA-seq of Tgfbr2-depleted tendons showed that as the cells lose TGFβ, they also lose expression of tendon markers (Scx, Tnmd, etc.), increase their clonal potential, and gain expression of markers that have been reported as stem/progenitor markers (Sca-1, Cd34, and Cd44). However, the authors were unable to demonstrate that these genes were expressed by embryonic tendon progenitors within the tendon fascicle. This work putatively demonstrates the flexibility of tenocyte fate during development and growth, which could have implications for tendon pathology and regeneration. They then went on to rescue the phenotype within individual cells via viral expression of Tgfbr2, which is and interesting and elegant an innovative approach. This study is very thorough with innovative findings that will advance our understanding of tenogenic differentiation.

Essential revisions:

1) The amount of Tgfbr2 deletion in Scx-CreERt2 and RosaCreERt2 (which results in lack of a mutant phenotype) is critical data to argue the mutant phenotype is based the temporal nature of the Scx-Cre. These results imply that TGFβ signaling should active be in tendon cells during development and postnatal stages. Do we know the endogenous pattern of TGFβ signaling activity in tendons until P15? Is it possible to define if there is a specific stage when the presence of TGFβ signaling is critical to maintain tendon cell fate? This lack of phenotype generated a lot of discussion among the reviewers and it was not mentioned in the Materials and methods section when the tamoxifen injections were conducted, further confusing this issue. It was felt that is might be an interesting finding but that there must be proof showing that this was not a technical issue or a failure of the cre- to function. If a lack a technical issue explaining the lack of phenotype cannot ruled out, these data may need to be removed.

2) Related to the above issue, how efficient is the knockdown of Tgfbr2 in Scx-Cre? This would be an important comparison to confirm that the mutant phenotype is indeed time and lineage driven. In addition, understanding the degree of TgfBr2 knockdown is important as it is very likely that the specific effects of Tgfb signaling are highly dose dependent, that is, there is clearly a level of Tgfb signaling that is required to maintain tenocyte fate, but Tgfb is also a potent inducer of myofibroblast differentiation and fibrosis, which complicates the use of the Tgfb as a therapeutic.

3) The increased clonogenicity helps support the conclusion of dedifferentiation and the genes that are enriched in these cells (Sca-1, Cd34, Cd44, Dpt, Anxa1, CD34, CD44, Mgp and Mfap5) have been reported as general stem/progenitor markers or are expressed during early tendon development. The challenge with this conclusion, which the authors acknowledge, is that our understanding of markers at different stages of the tendon lineage is incomplete, which makes it even more difficult to prove that dedifferentiation occurs. As the authors noted and presented in Figure 6B, several of these genes are elevated in healing as well. The disintegration of the tendon likely elicits a healing response, which would almost certainly alter the cell phenotype. In addition, the phenotype is only seen with the ScxCre and not the CreERT2 models, suggesting that a large proportion of the tendon cell population needs to be modulated to elicit enough alterations in ECM and concurrent cell dedifferentiation. By the time the dedifferentiation occurs the ECM is already severely disrupted. Can the authors please discuss/defend their conclusion that this is dedifferentiation and not a general healing response?

4) The fact that the phenotype doesn't manifest until postnatal ages suggests that a mechanical loading threshold may exist that tips the scales towards disintegration of the tendon. Discussion of mechanics is warranted and would greatly benefit the manuscript.

5) Figure 4B- the authors state that the enlarged Scx-GFP cells in the mutant are newly recruited tendon cells- can they clarify what they mean by this? Are these cells from the tendon that actively express Scx-GFP or extrinsic Scx-expressing cells that migrate to the tendon. The reviewers respected that this is the basis for another manuscript but felt that this is very striking finding and an important statement that must be clarified.

6) Figure 4D- The authors trace with Scx-Cre; RosaAi9 to 'prove' that these green cells are tendon derived. However, there is evidence from the authors and other groups identifying non-tendon targeting with this cre, including muscle connective tissue cells, and bone (Mckenzie et al., 2017). A tendon graft may be the only way to 'prove' these are tendon derived. Given that these experiments are not feasible, the reviewers suggested a slight modification of this text to indicate that these data suggest these are tendon derived cells.

7) Do the authors know what percent of mutant cells were successfully targeted by the viral infection? That is, have they looked at V5 expression in Scx-Cre; RosaTd+ mutants since they say in subsection “Tenocyte dedifferentiation is dependent on cell autonomous loss of TGFβ signalling” that infected cells will be surrounded by mutant cells.

8) Is the data presented in Figure 7D from injection with the Cre dependent AAV or the constitutive Tgfbr2 construct? This is unclear from the Results section. If the P1 data are with the constitutive Tgfbr2 construct they should be plotted separately.

9) The authors performed single-cell RNA-seq and at the end only show differentially expressed genes that could have been obtained with bulk RNA-seq. It was strongly felt by all reviewers that the authors should exploit the scRNAseq data more fully to support their conclusions. Further, it was unanimously felt that the authors should show the clustering of both normal tendons and TGFBRII-depleted tendons. Researchers working in tendon area have been eagerly awaiting scRNAseq data of normal tendon to identify the different cell populations in tendon since tendon fibroblasts are uncharacterized. Further, the clustering of TGFBRII-depleted tendons will be very informative to determine which tendon cell types are affected in this mutant condition. An exhaustive analysis was not requested, and it was not felt that this should not take too long to do. Clustering with the Cell Rangers 10X software is immediate. A bioinformatics analysis could place the paper in a very attractive position which would he highly advantageous to the authors.

---

## [Author Response]

Summary:This study continues years of work by this lab studying the role TGFβ signaling in tendon differentiation. Previously this group demonstrated that TGFβ is required for tendon specification and formation (Pryce, 2009). In the current study, they used ScxCre mice to target tendon cells after specification and established that TGFβ signaling needs to be maintained by tendon cells to continue their differentiation and preserve a normal ECM. Interestingly, the cell phenotype and ECM does not appear to be abnormal until approximately 7 days after birth. After this stage, the tendons begin to disintegrate and the mutant mice actually die before P14. Analysis of single-cell RNA-seq of Tgfbr2-depleted tendons showed that as the cells lose TGFβ, they also lose expression of tendon markers (Scx, Tnmd, etc.), increase their clonal potential, and gain expression of markers that have been reported as stem/progenitor markers (Sca-1, Cd34, and Cd44). However, the authors were unable to demonstrate that these genes were expressed by embryonic tendon progenitors within the tendon fascicle. This work putatively demonstrates the flexibility of tenocyte fate during development and growth, which could have implications for tendon pathology and regeneration. They then went on to rescue the phenotype within individual cells via viral expression of Tgfbr2, which is and interesting and elegant an innovative approach. This study is very thorough with innovative findings that will advance our understanding of tenogenic differentiation.Essential revisions:1) The amount of Tgfbr2 deletion in Scx-CreERt2 and RosaCreERt2 (which results in lack of a mutant phenotype) is critical data to argue the mutant phenotype is based the temporal nature of the Scx-Cre. These results imply that TGFβ signaling should active be in tendon cells during development and postnatal stages. Do we know the endogenous pattern of TGFβ signaling activity in tendons until P15? Is it possible to define if there is a specific stage when the presence of TGFβ signaling is critical to maintain tendon cell fate? This lack of phenotype generated a lot of discussion among the reviewers and it was not mentioned in the Materials and methods section when the tamoxifen injections were conducted, further confusing this issue. It was felt that is might be an interesting finding but that there must be proof showing that this was not a technical issue or a failure of the cre- to function. If a lack a technical issue explaining the lack of phenotype cannot ruled out, these data may need to be removed.

We agree with the reviewers that pattern of TGFβ signaling activity and the efficiency of Cre activity in the inducible experiments are critical for a complete analysis of the phenotype.

The pattern of TGFβ signaling is usually detected by following the phosphorylation of Smad2 and has been of great importance both for our previous studies regarding TGFβ signaling and for this project. Unfortunately, despite numerous attempts to perform pSmad2 antibody staining to detect endogenous pattern of TGFβ signaling we could not achieve a reliable staining and therefore, cannot establish the pattern of TGFβ signaling.

For the inducible Cre studies, the existing *ScxCreERT2* tendon inducible cre is active only in post-natal stages and so far there is no good inducible Cre to target tendons at embryonic stages. A systematic developmental targeting of the TGFβ type II receptor was therefore not feasible, precluding the ability to define if there is a specific stage when the presence of TGFβ signaling is critical to maintain the tendon cell fate.

Since the cellular phenotype manifested only in early post-natal stages, we performed the post-natal deletion of *Tgfbr2* using both *ScxCreCreERT2* and *RosaCreERT2* and detected no loss of tendon markers. As requested, a more detailed description of the experiment and the specific timing and dose of tamoxifen injections were now added to the text (subsection “Loss of the tendon cell fate in mutant tenocytes”). Loss of the receptor in these experiments was verified using antibody staining for TGFβ type II receptor and the results are shown as a supplementary figure (Figure 1—figure supplement 1D,E).

While we recognize that these results do not provide an adequate developmental targeting of the receptor we feel that it is of great importance to include them in the text because they demonstrate that simple postnatal loss of TGFβ signaling is not sufficient to induce tenocyte dedifferentiation, suggesting that dedifferentiation involves a more complex process and also indicating to other researchers who may be interested to introduce this effect in their experimental systems that they cannot expect to achieve tenocyte dedifferentiation with a simple targeting of the receptor.

2) Related to the above issue, how efficient is the knockdown of Tgfbr2 in Scx-Cre? This would be an important comparison to confirm that the mutant phenotype is indeed time and lineage driven. In addition, understanding the degree of TgfBr2 knockdown is important as it is very likely that the specific effects of Tgfb signaling are highly dose dependent, that is, there is clearly a level of Tgfb signaling that is required to maintain tenocyte fate, but Tgfb is also a potent inducer of myofibroblast differentiation and fibrosis, which complicates the use of the Tgfb as a therapeutic.

Thank you for pointing this out. Our experience using Cre reporters is that *ScxCre* activity within tendons is not uniform during embryogenesis. However, the percentage of targeted tendon cells increases during development and reaches close to 100% [based on *Ai14 Rosa26-tdTomato (RosaT)* Cre reporter expression] at P0.

To directly determine the efficiency of *Tgfbr2* knockdown in *Tgfbr2;ScxCre* mutants we performed immunostaining of TGFβ type II receptor in P0 samples and found complete loss of the receptor in tendons. The images have been added as a supplementary figure (Figure 1—figure supplement 1B,C). The manuscript text has also been revised accordingly (subsection “Targeting TGFβ type II receptor in Scx*-*expressing cells resulted in tendon disruption and limb abduction”).

For the second comment regarding levels of TGFβ signaling and therapeutic relevance. As noted above we cannot directly determine the changes in the levels of TGFβ signaling we certainly agree that our results should not be interpreted in the context of using TGFβ as a therapeutic agent and have not made any such inferences in the text.

3) The increased clonogenicity helps support the conclusion of dedifferentiation and the genes that are enriched in these cells (Sca-1, Cd34, Cd44, Dpt, Anxa1, CD34, CD44, Mgp and Mfap5) have been reported as general stem/progenitor markers or are expressed during early tendon development. The challenge with this conclusion, which the authors acknowledge, is that our understanding of markers at different stages of the tendon lineage is incomplete, which makes it even more difficult to prove that dedifferentiation occurs. As the authors noted and presented in Figure 6B, several of these genes are elevated in healing as well. The disintegration of the tendon likely elicits a healing response, which would almost certainly alter the cell phenotype. In addition, the phenotype is only seen with the ScxCre and not the CreERT2 models, suggesting that a large proportion of the tendon cell population needs to be modulated to elicit enough alterations in ECM and concurrent cell dedifferentiation. By the time the dedifferentiation occurs the ECM is already severely disrupted. Can the authors please discuss/defend their conclusion that this is dedifferentiation and not a general healing response?

The cell fate changes begin at P2 and actually precede the matrix disruption that is not observed before P7 (compare Figure 1—figure supplement 2 and Figure 3). We agree with the reviewers that this is a critical point for understanding the key drivers of tenocyte dedifferentiation in these mutants and therefore revised text to explicitly highlight this point (subsection “Targeting TGFβ type II receptor in Scx*-*expressing cells resulted in tendon disruption and limb abduction”). In respect to the inducible experiments, as noted above the loss of the receptor is nearly complete in the inducible mutants (Figure 1—figure supplement 1D,E), so the differences between the models are not due to extent of receptor targeting.

In regards to the issues of dedifferentiation vs. healing response, we agree and have noted in the text that mutant tenocytes did not turn on embryonic tendon progenitor genes, rather that they express more general progenitor markers. We believe and have now redoubled our efforts to explicitly discuss (subsection “Mutant tenocytes acquired stem/progenitor features”, Discussion section) that this kind of change in cell fate is also compatible with dedifferentiation and should be recognized as such. Moreover, since the identity and expression profile of tendon stem/progenitor cells that contribute to tendon healing has not been established so far it may well be possible that the cells will share more similarities with the dedifferentiated tenocytes in our mutants.

We further very much agree that the expression profile at P7 possibly represents a healing response. Notably it is usually suggested or implied that the cells with this expression profile are cells that were recruited to an injury site within the healing response. The important and it is the resident novel concept in this mutant analysis is that the resident tenocytes de-differentiate and they are the ones expressing these genes and assuming enhanced clonogenicity. This is also more explicitly stated in the Discussion section.

4) The fact that the phenotype doesn't manifest until postnatal ages suggests that a mechanical loading threshold may exist that tips the scales towards disintegration of the tendon. Discussion of mechanics is warranted and would greatly benefit the manuscript.

Thanks for the suggestion. Discussion of mechanics has been added into the revised manuscript (Discussion section).

5) Figure 4B- the authors state that the enlarged Scx-GFP cells in the mutant are newly recruited tendon cells- can they clarify what they mean by this? Are these cells from the tendon that actively express Scx-GFP or extrinsic Scx-expressing cells that migrate to the tendon. The reviewers respected that this is the basis for another manuscript but felt that this is very striking finding and an important statement that must be clarified.

The finding of these cells are recruited is based mainly on the fact that some of these *ScxGFP*-positive cells exhibited weak or no expression of the *RosaT* Cre reporter. By P0 the cre reporter is positive in all tendon cells of wild-type tendons. The absence of Cre reporter thus indicates a recent activation of the *Scx* enhancer in these cells. This finding was corroborated with other experiments including the double-fluorescent cre reporter mTmG. Subsection “Loss of the tendon cell fate in mutant tenocytes” has been revised to clarify this part. We agree that this is a very exciting finding; however, it is impossible to include the full description of the finding in this study. The manuscript is currently under preparation and will be submitted for publication soon.

6) Figure 4D- The authors trace with Scx-Cre; RosaAi9 to 'prove' that these green cells are tendon derived. However, there is evidence from the authors and other groups identifying non-tendon targeting with this cre, including muscle connective tissue cells, and bone (Mckenzie et al., 2017). A tendon graft may be the only way to 'prove' these are tendon derived. Given that these experiments are not feasible, the reviewers suggested a slight modification of this text to indicate that these data suggest these are tendon derived cells.

First a clarification, Figure 4D shows that the *ScxGFP*-negative cells, i.e. NOT green cells (*ScxGFP*-positive), in mutant tendons were tendon-derived based on the *RosaT* Cre reporter.

We agree that *ScxCre* labels also non-tendon cells and therefore that the mere fact that cells in mutant tendons are reporter-positive is not sufficient to indicate that these are tendon-derived cells. We therefore combined another rationale to address this issue. At P7, 75-80% of cells within all mutant tendons are both *ScxGFP*-negative and *RosaT*-positive (with the remaining are the recruited cells). Given that there is no apparent elimination of the existing tendon cells (i.e. no cell death), repopulation of the tendon by neighboring reporter positive cells would not explain the absence of cells that express tenocyte markers in these tendons. We thus suggest that the only logical explanation to these observations is that the dedifferentiated cells within the mutant tendon are the original tendon cells, i.e. tendon-derived cells (subsection “Loss of the tendon cell fate in mutant tenocytes”).

7) Do the authors know what percent of mutant cells were successfully targeted by the viral infection? That is, have they looked at V5 expression in Scx-Cre; RosaTd+ mutants since they say in subsection “Tenocyte dedifferentiation is dependent on cell autonomous loss of TGFβ signalling” that infected cells will be surrounded by mutant cells.

The efficiency of virus injections is highly variable in our experiments and quantification of infection efficiency may not be informative. However, our experience with AAV infection is that while muscle infection is robust (e.g. Huang et al., 2013) tenocyte infection is not efficient and therefore always results in sporadic and low infection (<10%). Therefore, we usually observed one or two individual infected cells surrounded by mutant cells, as also shown in Figure 7—figure supplement 1.

8) Is the data presented in Figure 7D from injection with the Cre dependent AAV or the constitutive Tgfbr2 construct? This is unclear from the Results section. If the P1 data are with the constitutive Tgfbr2 construct they should be plotted separately.

The reviewers are correct in stating that the data from embryonic injections are with a cre dependent virus while for P1 injection was of the constitutive *Tgfbr2* AAV virus. The rationale behind this experimental design was the interest to test the outcome of maintenance of *Tgfbr2* expression in isolated tenocytes for embryonic experiments that could use the cre activation scheme and of re-expression of TGFβ type II receptor in dedifferentiated cells that are therefore likely not *ScxCre* positive and therefore we used a ubiquitous expressor (see the Materials and Method section and subsection “Tenocyte dedifferentiation is dependent on cell autonomous loss of TGFβ signaling”). The main purpose of this data (Figure 7D) is to show that re-expression of *Tgfbr2* at the designated stages is sufficient to prevent or rescue the loss of tendon cell fate, irrespective of the stage in which the receptor was reintroduced. By presenting data in one plot, we find it is much easier and more compelling to convey the message that reintroduction of the receptor is sufficient to reactivate the tendon cell fate irrespective of stage. Since only cells within tendons were evaluated for this data the difference in virus construction was not pertinent to the outcome for the cells. For clarity we added however a note of this different virus structure to the Figure 7 legend.

9) The authors performed single-cell RNA-seq and at the end only show differentially expressed genes that could have been obtained with bulk RNA-seq. It was strongly felt by all reviewers that the authors should exploit the scRNAseq data more fully to support their conclusions. Further, it was unanimously felt that the authors should show the clustering of both normal tendons and TGFBRII-depleted tendons. Researchers working in tendon area have been eagerly awaiting scRNAseq data of normal tendon to identify the different cell populations in tendon since tendon fibroblasts are uncharacterized. Further, the clustering of TGFBRII-depleted tendons will be very informative to determine which tendon cell types are affected in this mutant condition. An exhaustive analysis was not requested, and it was not felt that this should not take too long to do. Clustering with the Cell Rangers 10X software is immediate. A bioinformatics analysis could place the paper in a very attractive position which would he highly advantageous to the authors.

Our lab is funded to perform a comprehensive scRNASeq analysis of tendons from embryonic stages and up to adult mice. We recognize the urgency for the field to have reliable single cell data and have therefore expedited analysis of one stage that will be ready hopefully within a few months. Through these studies we however learned to appreciate the complexity of these analyses and the fact that clustering is not a simple objective result and it can be different by changing algorithm or parameters and that it is therefore advisable to publish “an atlas” describing the cells in a tissue only after careful examination and validation of the cell types that manifest. In a counterintuitive fashion WT analysis requires more validation and confirmation than that of a mutant analysis since in the mutant it is possible to simply focus on comparing one set of clusters and the focus is on the difference between such clusters irrespective of the definitions and identities of all the other clusters. We agree with the reviewers that this is similar to the results usually associated with bulk analysis which would not have been possible in these mutants due to the cell complexity of dedifferentiated and newly recruited cells. We therefore preferred to avoid a comprehensive discussion of the wild-type data – that may if it is too detailed also detract from the flow of logic in the manuscript.

However recognizing the need that was reflected in the reviewers’ comments t-SNE plots and a list of top signature genes for the normal and mutant tendons have been added (Figure 6A, Supplementary file 1).

Please also note that scRNASeq data for both P7 mutant and wild-type pups has been deposited onto GEO repository for open access to readers

(https://www.ncbi.nlm.nih.gov/geo/query/acc.cgi?acc=GSE139558; token to access the data: elkhqeqmppgpbgl).